# Nuclear Argonaute protein NRDE-3 switches small RNA partners during embryogenesis to mediate temporal-specific gene regulatory activity

**Shihui Chen, Carolyn Marie Phillips***

Department of Biological Sciences, University of Southern California, Los Angeles, United States

## eLife Assessment

The study by Chen and Phillips provides evidence for a dynamic switch in the small RNA repertoire of the Argonaute protein NRDE-3 during embryogenesis in *C. elegans*. The work is supported by **convincing** experimental data, shedding light on RNA regulation during development. While the functional relevance of this process warrants further investigation, this study provides **valuable** insights into small RNA pathways with broader implications for developmental biology and gene regulation in other systems.

**\*For correspondence:**
cphil@usc.edu

**Competing interest:** The authors declare that no competing interests exist.

**Abstract** RNA interference (RNAi) is a conserved pathway that utilizes Argonaute proteins and their associated small RNAs to exert gene regulatory function on complementary transcripts. While the majority of germline-expressed RNAi proteins reside in perinuclear germ granules, it is unknown whether and how RNAi pathways are spatially organized in other cell types. Here, we find that the small RNA biogenesis machinery is spatially and temporally organized during *Caenorhabditis elegans* embryogenesis. Specifically, the RNAi factor, SIMR-1, forms visible concentrates during mid-embryogenesis that contain an RNA-dependent RNA polymerase, a poly-UG polymerase, and the unloaded nuclear Argonaute protein, NRDE-3. Curiously, coincident with the appearance of the SIMR granules, the small RNAs bound to NRDE-3 switch from predominantly CSR-class 22G-RNAs to ERGO-dependent 22G-RNAs. NRDE-3 binds ERGO-dependent 22G-RNAs in the somatic cells of larvae and adults to silence ERGO-target genes; here we further demonstrate that NRDE-3-bound, CSR-class 22G-RNAs repress transcription in oocytes. Thus, our study defines two separable roles for NRDE-3, targeting germline-expressed genes during oogenesis to promote global transcriptional repression, and switching during embryogenesis to repress recently duplicated genes and retrotransposons in somatic cells, highlighting the plasticity of Argonaute proteins and the need for more precise temporal characterization of Argonaute-small RNA interactions.

## Introduction

Precise gene expression is essential for organisms at all developmental stages. Small RNAs and their partners, the Argonaute (AGO) proteins, play an important role in regulating gene expression by targeting and silencing complementary nucleic acid sequences. This small RNA-mediated gene silencing process is known as RNA interference (RNAi) (*Fire et al., 1998*). The nematode *Caenorhabditis elegans*, distinguished by its expanded Argonaute family and intricate RNAi pathway, is a well-established model organism to study the RNAi pathway. *C. elegans* has 19 functional Argonaute

proteins and various classes of small RNAs (*Yigit et al., 2006*; *Seroussi et al., 2023*), which is greatly expanded compared to 8 Argonaute proteins in mammals, 5 in *Drosophila melanogaster*, and 1 in *Schizosaccharomyces pombe* (*Höck and Meister, 2008*). This expansion of the Argonaute family in nematodes may be linked to the diversity of habitats in which nematodes reside and environmental cues to which they must respond. First, RNAi has been well-studied in plants for its role as an antiviral defense mechanism (*Ding et al., 2004*); and like plants, worms lack an adaptive immune system, making the RNAi system a primary means to respond to viral intruders (*Félix et al., 2011*; *Ashe et al., 2013*; *Sarkies and Miska, 2013*). Second, nematodes have a specialized nucleic acid transporter required for the uptake of double-strand (ds)RNA from the intestinal lumen (*McEwan et al., 2012*; *Winston et al., 2007*), indicating that environmental sensing mediated by ingested dsRNA is an important aspect of nematode physiology (*Sarkies and Miska, 2013*). Lastly, it has been proposed that Ago diversity and rapid evolution could be linked to the environmental plasticity of nematodes, including the capacity for parasitism and challenges of invading and colonizing a host (*Buck and Blaxter, 2013*). Regardless of the evolutionary origin for the expansion of RNAi pathway proteins in nematodes, these pathways are not only important for a response to the environment, but are essential for the regulation of thousands of endogenous genes. Therefore, untangling the details of RNA silencing in *C. elegans* will shed light on the mechanisms of small RNA-mediated gene regulation in *C. elegans* and other organisms.

Argonaute proteins can be subdivided into three clades. Proteins are grouped into the AGO and PIWI clades based on their similarity to *Arabidopsis thaliana* AGO1 and *Drosophila melanogaster* PIWI, respectively. The third, WAGO, clade represents a nematode-specific expansion of the Argonaute protein family (*Yigit et al., 2006*). While small RNAs bound by the AGO- and PIWI-clade Argonaute proteins tend to be processed from longer, precursor transcripts, the WAGO-clade Argonaute proteins bind 22-nucleotide, 5'-triphosphorylated small RNAs (22G-RNAs) with which are each de novo synthesized by RNA-dependent RNA polymerases (RdRPs) (*Gu et al., 2009*; *Pak and Fire, 2007*; *Aoki et al., 2007*). However, even within the WAGO clade, each of the 11 Argonaute proteins exhibits specificity for a unique group of 22G-RNAs and exhibits distinct tissue and developmental expression patterns (*Seroussi et al., 2023*). For example, WAGO-1 binds 22G-RNAs that target transposons, pseudogenes, and aberrant transcripts, and silences genes post-transcriptionally in the germline cytoplasm (*Gu et al., 2009*), while CSR-1 binds 22G-RNAs targeting germline-expressed genes, functioning to clear maternal mRNA in early embryos while licensing and tuning gene expression in the adult germline (*Quarato et al., 2021*; *Gu et al., 2009*; *Claycomb et al., 2009*). Other WAGO Argonautes, such as SAGO-1 and SAGO-2, function exclusively in somatic cells and play roles in regulating endogenous genes, exogenous RNAi, and immunity (*Seroussi et al., 2023*). Unique amongst the WAGO Argonautes for their nuclear localization are HRDE-1 and NRDE-3, which are thought to silence genes co-transcriptionally in germline and soma respectively, and are required for the inheritance of RNA silencing signals from parents to offspring (*Buckley et al., 2012*; *Guang et al., 2008*). Despite extensive characterization of the *C. elegans* Argonaute proteins, we still know little about the factors that promote the spatiotemporal expression of each Argonaute protein and the mechanisms that promote Argonaute-small RNA binding specificity. Furthermore, most Argonaute-small RNA sequencing experiments have been performed at a single time point, usually in adult *C. elegans*, meaning that we have little understanding as to how the RNA targets of each Argonaute protein change across development.

In the *C. elegans* germline, many of the RNAi components, including Argonaute proteins, RdRPs, and other small RNA processing machinery, localize within phase-separated germ granules. Often, proteins acting in different functional branches of the RNAi pathway seem to reside in separate compartments of the germ granules, suggesting that there are specialized areas within the germ granules where distinct molecular reactions occur. Presently, six sub-compartments of the germ granule have been identified in *C. elegans*: P granules, Z granules, *Mutator* foci, SIMR foci, and more recently, E granules and D granules (*Brangwynne et al., 2009*; *Phillips et al., 2012*; *Wan et al., 2018*; *Manage et al., 2020*; *Chen et al., 2024c*; *Huang et al., 2024*). Processing bodies (P-bodies), a well-characterized condensate that contains proteins associated with mRNA turnover and translationally repressed mRNAs, also closely associate with germ granules (*Du et al., 2023*). These germ granule compartments are situated at the cytoplasmic side of the nucleus, proximal to nuclear pores. However, the mechanisms governing their spatial organization remain unknown. Moreover, with the

majority of studies focusing on mechanisms of RNA silencing and germ granule organization in the germline, there is limited understanding of how each of these germ granule compartments assembles and functions in embryos. It has been observed that in *C. elegans* embryogenesis, the primordial germline cell P4 divides into Z2 and Z3 progenitor germ cells (PGCs) at around the 100 cell stage, coinciding with the demixing of Z granules from P granules, the appearance of *Mutator* foci and SIMR foci, and the initiation of germ cell transcription (*Updike and Strome, 2010*; *Uebel et al., 2021*; *Wan et al., 2018*; *Seydoux and Dunn, 1997*). Together, the assembly of this more complex germ granule organization coinciding with a burst of transcription from the germ cells, may indicate that these multi-compartment structures are necessary to monitor the newly synthesized germline transcripts. Yet even these limited studies of RNAi pathway factors in embryos fail to address a role for ribonucleoprotein granules in RNA silencing in the soma.

Here, we discovered that SIMR-1, the founding component of the germline SIMR foci, is also found in cytoplasmic granules in the somatic cells of *C. elegans* embryos. These embryonic 'SIMR granules' additionally contain factors involved in 22G-RNA amplification and associated with the nuclear Argonaute protein, NRDE-3. However, NRDE-3 itself only associates with the SIMR granules when not bound to small RNAs. Strikingly, the SIMR granules exhibit temporal dynamics where they first appear in early embryogenesis (around the 8-cell stage), peak around the 100 cell stage, and have mostly disappeared by the comma stage of embryogenesis. Curiously, these embryonic SIMR granules are by no means the only RNAi-related embryonic granules, as numerous other RNAi factors are found in separate granules in embryos, including components of the CSR pathway, the Argonaute CSR-1 and its RdRP EGO-1. Interestingly, clearance of many of these granules, including the embryonic SIMR, CSR, Z and P granules, is regulated by a common mechanism of autophagic degradation. Furthermore, by sequencing the small RNAs bound by NRDE-3 in early and late embryogenesis, we found that the formation of the SIMR granules coincides with a switch in NRDE-3 small RNA targets, from CSR-class 22G-RNAs to ERGO-dependent 22G-RNAs. Together, our data demonstrates that NRDE-3 has two separate functions, first acting in oogenesis to repress RNA Polymerase II and promote genome-wide transcription repression, and second acting downstream of ERGO-1 to transcriptionally silence retrotransposons, pseudogenes, and aberrant transcripts. Further, the SIMR granules themselves appear to be sites of NRDE-3-bound 22G-RNA biogenesis and loading and may contribute to the efficiency or specificity of Argonaute-small RNA interactions during embryogenesis.

## Results

### SIMR-1 and ENRI-2 localize to cytoplasmic granules during embryogenesis

In previous work, we sought to identify proteins that associate with SIMR-1 and ultimately found that SIMR-1 associates with HRDE-2 and the nuclear Argonaute protein, HRDE-1, to promote correct HRDE-1 small RNA binding in germ cells (*Chen and Phillips, 2024a*). In that work, we also identified another nuclear Argonaute protein, NRDE-3, as an interactor of SIMR-1. NRDE-3 has been shown previously to bind ERGO-dependent 22G-RNAs and silence ERGO-target genes in somatic cells (*Figure 1—figure supplement 1* A-B). To delve further into this potential interaction between SIMR-1 and NRDE-3, we first systematically compiled a list of protein interactions identified from previous studies for both SIMR-1 and NRDE-3 (*Figure 1A*). Interestingly, the HRDE-2 paralog, ENRI-2, had been shown to interact with both SIMR-1 and NRDE-3 in embryos by immunoprecipitation (IP) followed by mass spectrometry (mass-spec), and another HRDE-2 paralog, ENRI-1, was similarly shown to interact with only NRDE-3 (*Lewis et al., 2020*). These findings suggest that SIMR-1, NRDE-3, ENRI-2, and possibly ENRI-1 proteins may function together in the somatic nuclear RNAi pathway, analogous to the roles of SIMR-1, HRDE-1, and HRDE-2 in the germline nuclear RNAi pathway.

Here, we first aimed to address whether and where NRDE-3, SIMR-1, ENRI-1, and ENRI-2 colocalize. NRDE-3 has previously been shown to be expressed in the nucleus of most somatic cells (*Guang et al., 2008*). Until recently, all characterization of NRDE-3 was done using a low-copy, integrated transgenic strain in which the nuclear localization was not visible until the ~30–80 cell stage of development, and it was presumed that this localization reflected the endogenous NRDE-3 localization (*Guang et al., 2008*; *Lewis et al., 2020*). However, a more recent study constructed an endogenously-tagged NRDE-3 strain using CRISPR and found that NRDE-3 additionally localizes to the nucleus of oocytes

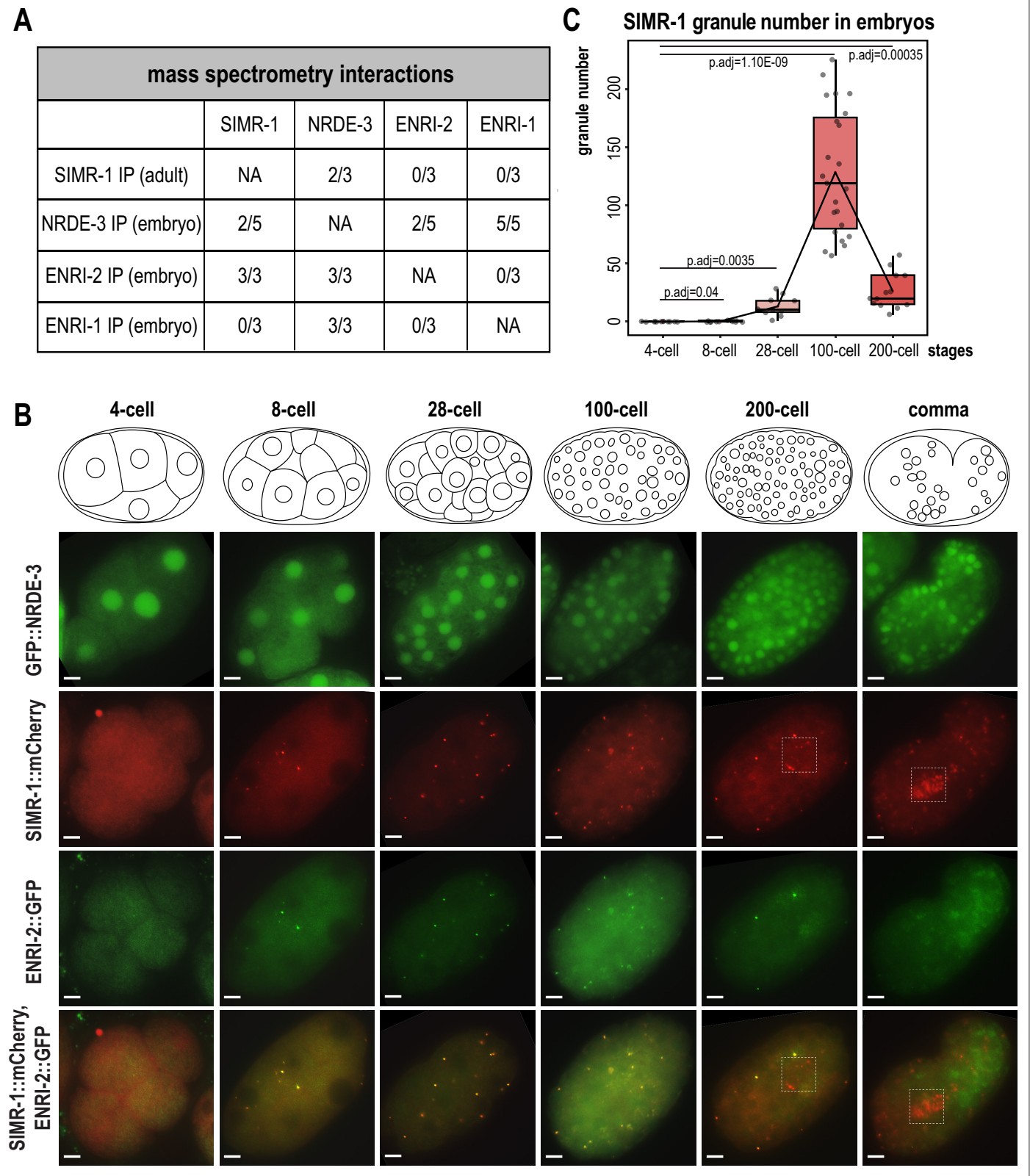

**Figure 1.** SIMR-1 and ENRI-2 colocalize at somatic granules in embryos. (**A**) Summary of IP-mass spectrometry interactions detected between NRDE-3, ENRI-2, ENRI-1, and SIMR-1 from previously published studies (***Chen and Phillips, 2024a***; ***Lewis et al., 2020***). The number of replicates from which the interaction was detected relative to the total number of replicates performed is indicated. (**B**) Live imaging of GFP::3xFLAG::NRDE-3 and SIMR-1::mCherry::2xHA; ENRI-2::2xTy1::GFP embryos at different stages (4 cell, 8 cell, 28 cell, 100 cell, 200 cell, and comma). Boxes identify the location of $Z_2$

*Figure 1 continued on next page*

*Figure 1 continued*

and $Z_3$ primordial germ cells, showing that SIMR-1 is present in germ granules while ENRI-2 is not. At least five individual embryos were imaged for each genotype and stage. Scale bars, 5 µm. (**C**) Box plot of SIMR-1::mCherry::2xHA granule number quantification at different embryonic stages (4 cell, 8 cell, 28 cell, 100 cell, and 200 cell). At least ten individual embryos at each stage were used for quantification. Each dot represents an individual embryo, and all data points are shown. Bolded midline indicates median value, box indicates the first and third quartiles, and whiskers represent the most extreme data points within 1.5 times the interquartile range. Lines connect the mean granule number for each stage, illustrating the change in number of SIMR granules across the developmental stages of the embryo. Two-tailed t-tests were performed to determine statistical significance and p-values were adjusted for multiple comparisons. See Materials and methods for a detailed description of quantification methods.

The online version of this article includes the following source data and figure supplement(s) for figure 1:

**Source data 1.** This file contains the raw data used to generate the graph of SIMR granule number shown in *Figure 1C*.

**Figure supplement 1.** Summary of NRDE-3 and CSR-1 small RNA pathway and components.

**Figure supplement 2.** Expression of NRDE-3, SIMR-1, and ENRI-1.

**Figure supplement 2—source data 1.** This file contains the original western blots of ENRI-1::mCherry::2xHA used in creating *Figure 1—figure supplement 2D*, indicating the relevant bands, antibodies, strains, and developmental stages.

**Figure supplement 2—source data 2.** This folder contains the original files for western blot analysis displayed in *Figure 1—figure supplement 2D*.

and early embryos (*Seroussi et al., 2023*; *Seroussi et al., 2022*). In a recent study, we addressed this discrepancy, and demonstrated that the older, transgenic NRDE-3 construct is targeted for RNAi-mediated gene silencing in germ cells and early embryos (*Chen and Phillips, 2024b*). SIMR-1 is a component of the SIMR foci, a sub-compartment of germ granules, that appears as punctate foci at the periphery of *C. elegans* germ cells starting in embryos through the adult stage (*Manage et al., 2020*; *Uebel et al., 2021*). ENRI-1 has been reported to localize to the cytoplasm of oocytes and embryos while ENRI-2 localized to both the nucleus and cytoplasm, varying depending on developmental stage (*Lewis et al., 2020*). With these four proteins showing distinct localization patterns from one another, it was unclear how these proteins could physically interact.

To investigate where and how these interactions might potentially occur, we chose to initially examine localization of these proteins in the germline of adult *C. elegans* using the endogenously-tagged NRDE-3 strain which is visible starting in late pachytene. As expected based on previous work, NRDE-3 localizes to the nucleus of germ cells, while SIMR-1 is found in the cytoplasm in SIMR foci, a compartment of the germ granule (*Figure 1—figure supplement 2A*; *Seroussi et al., 2022*; *Seroussi et al., 2023*; *Manage et al., 2020*). Next, we decided to examine NRDE-3 and SIMR-1 localization in embryos, carefully dividing the embryos into distinct developmental stages, from 4 cell to comma stage. As expected, we found that NRDE-3 is consistently localized to the nucleus in all embryonic stages (*Figure 1B*). Interestingly, we observed that SIMR-1 forms granules in the cytoplasm of somatic cells during some embryonic stages (*Figure 1B*). By quantifying the total number of granules per embryo across embryonic development, we found that the 'SIMR granules' first appear around the 8 cell stage and reach a peak at approximately the 100 cell stage, coinciding with the division of the germline precursor cell $P_4$ into the primordial germ cells $Z_2$ and $Z_3$ (*Wang and Seydoux, 2013*). Subsequently, the number of SIMR granules decreases, and in late embryos, SIMR-1 localizes primarily to the germ granules surrounding the two germ cells, as previously observed (*Uebel et al., 2021*; *Figure 1B–C*). We had previously shown that the Tudor domain of SIMR-1 was important for its assembly into germline SIMR foci. Therefore, we next explored the requirement for the Tudor domain in assembling SIMR-1 cytoplasmic granules in embryos (*Manage et al., 2020*). We found that the Tudor domain mutant, SIMR-1(R159C), fails to assemble in cytoplasmic granules in the embryos (*Figure 1—figure supplement 2B*), indicating that, similar to germline SIMR foci, the Tudor domain is also required for assembly of the cytoplasmic SIMR granules in embryos.

We next focused on ENRI-1 and ENRI-2 and observed that ENRI-2 shows similar cytoplasmic granule localization and colocalizes with SIMR-1 in embryos, but it does not localize to the germ granules in $Z_2$ and $Z_3$, suggesting that the activity of ENRI-2 is restricted to somatic cells (*Figure 1B*). Finally, we examined the localization of N-terminal tagged 2xTy1::GFP::ENRI-1, and could not detect any specific localization in either nuclei or cytoplasmic granules (*Figure 1—figure supplement 2C*; *Lewis et al., 2020*). Consequently, we constructed a new strain with C terminal tagged ENRI-1::mCherry::2xHA and confirmed the presence of full-length ENRI-1 protein with western blot (*Figure 1—figure supplement 2D*). Nonetheless, we could not detect any ENRI-1 localization in embryos with our newly generated

strain (*Figure 1—figure supplement 2C*). These results are consistent with the fact that ENRI-1 does not interact directly with either SIMR-1 or ENRI-2 by immunoprecipitation (*Figure 1A*; *Lewis et al., 2020*). Altogether, these data indicate that SIMR-1 and ENRI-2 colocalize at cytoplasmic granules in the somatic cells of embryos and suggest that ENRI-2 and SIMR-1 may function together at these sites. In contrast, NRDE-3 is spatially separated in the nucleus and no clear expression pattern was observed for ENRI-1.

## Unloaded NRDE-3 associates with SIMR-1 in cytoplasmic granules

Next, to determine whether SIMR-1 and ENRI-2 are required for NRDE-3 localization, we introduced the *simr-1* mutant, *enri-2* mutant, *enri-1* mutant, and *enri-1; enri-2* double mutant in the endogenously tagged GFP::3xFLAG::NRDE-3 strain, and examined NRDE-3 localization across embryonic developmental stages. We observed no changes in NRDE-3 expression or nuclear localization in any of the mutants examined at any developmental stage (*Figure 2—figure supplement 1A*).

In previous work, we demonstrated that the germline nuclear Argonaute protein HRDE-1 loses nuclear localization and associates in the cytoplasm with the SIMR compartment of germ granules when it is unable to bind small RNAs (*Chen and Phillips, 2024a*). Additionally, ENRI-2 interacts more strongly with NRDE-3 in an *eri-1* mutant background compared to wild-type (*Lewis et al., 2020*), suggesting that the interaction occurs when NRDE-3 does not bind small RNAs. Localization of unloaded NRDE-3 has been examined in the seam cells of L3 stage animals, where, like HRDE-1, it loses nuclear localization and becomes restricted to the cytoplasm (*Guang et al., 2008*). Therefore, we next sought to examine the localization of NRDE-3 when it is unbound to small RNA in embryos and germline. First, we aimed to deplete the preferred small RNA binding partners of NRDE-3. NRDE-3 has previously been shown to bind secondary 22G-RNAs downstream of ERGO-class 26G-RNAs, dependent on ERI-1, which is required for 26G-RNA biogenesis (*Guang et al., 2008*; *Han et al., 2009*; *Seroussi et al., 2023*), and RDE-3/MUT-2, which is a component of the *Mutator* complex and necessary for 22G-RNA production (*Chen et al., 2005*; *Phillips et al., 2012*; *Phillips et al., 2014*; *Figure 1—figure supplement 1A–B*). Therefore, we introduced an *eri-1* mutant and a *rde-3/mut-2* mutant into the endogenously GFP-tagged NRDE-3 background. We observed that NRDE-3 associates with somatic granules with a similar spatiotemporal pattern to SIMR-1 and ENRI-2, peaking around the 100 cell stage, although the total number of granules per embryo is lower for NRDE-3 granules in the *eri-1* and *rde-3/mut-2* mutant backgrounds compared to SIMR granules (*Figure 2A–B*). Next, to fully abolish the small RNA binding capacity of NRDE-3 and to confirm that the observed granule localization was due to the loss of small RNA loading, we introduced mutations to abolish small RNA binding into the GFP-tagged NRDE-3; specifically, residues 687 H and 691 K in the Mid domain were mutated to alanine, hereafter referred to as NRDE-3(HK-AA) (*Ma et al., 2005*; *Guang et al., 2008*; *Chen and Phillips, 2024a*). NRDE-3(HK-AA) localizes exclusively to the cytoplasm across embryonic development, accumulating in somatic granules at 100 cell stage similar to SIMR-1 and ENRI-2 (*Figure 2A*). Quantification of the number of NRDE-3 granules per embryo in the NRDE-3(HK-AA) strain shows that the dynamics of NRDE-3 granule appearance and disappearance are similar to that of SIMR-1, where the number of granules increases from early embryos up until about 100 cell stage and then decreases as the embryos progress to later stages of development (*Figure 2C*). Overall, the total number of NRDE-3(HK-AA) granules quantified per embryo are similar to or modestly higher than SIMR-1 granules (*Figures 1C and 2C*). Western blot analysis reveals that overall protein level are similar for wild-type NRDE-3 and NRDE-3(HK-AA) (*Figure 2—figure supplement 1B*), indicating that unloaded NRDE-3 is not subject to degradation. This result contrasts with our previous results for unloaded HRDE-1 (HK-AA), which does show reduced protein expression compared to wild-type HRDE-1 (*Chen and Phillips, 2024a*). Together, these data indicate that NRDE-3 forms granules in the cytoplasm of somatic cells when not associated with a small RNA binding partner.

It is also worth noting that despite the similarity in timing of NRDE-3 granule appearance and disappearance in the *eri-1* and *rde-3* mutants compared to the *nrde-3(HK-AA)* mutant, we observed a striking difference in the NRDE-3 localization in early embryos. Specifically, in *eri-1* and *rde-3* mutants, NRDE-3 localizes to the nucleus in early embryos while NRDE-3(HK-AA) localizes exclusively to the cytoplasm (*Figure 2A*). Similarly, in the Z2 and Z3 primordial germ cells of late embryos, NRDE-3 is still found in the nucleus in *eri-1* and *rde-3* mutants. In contrast, NRDE-3 localizes exclusively to the cytoplasm in the somatic cells of late embryos of all three mutants. This result suggests that there may

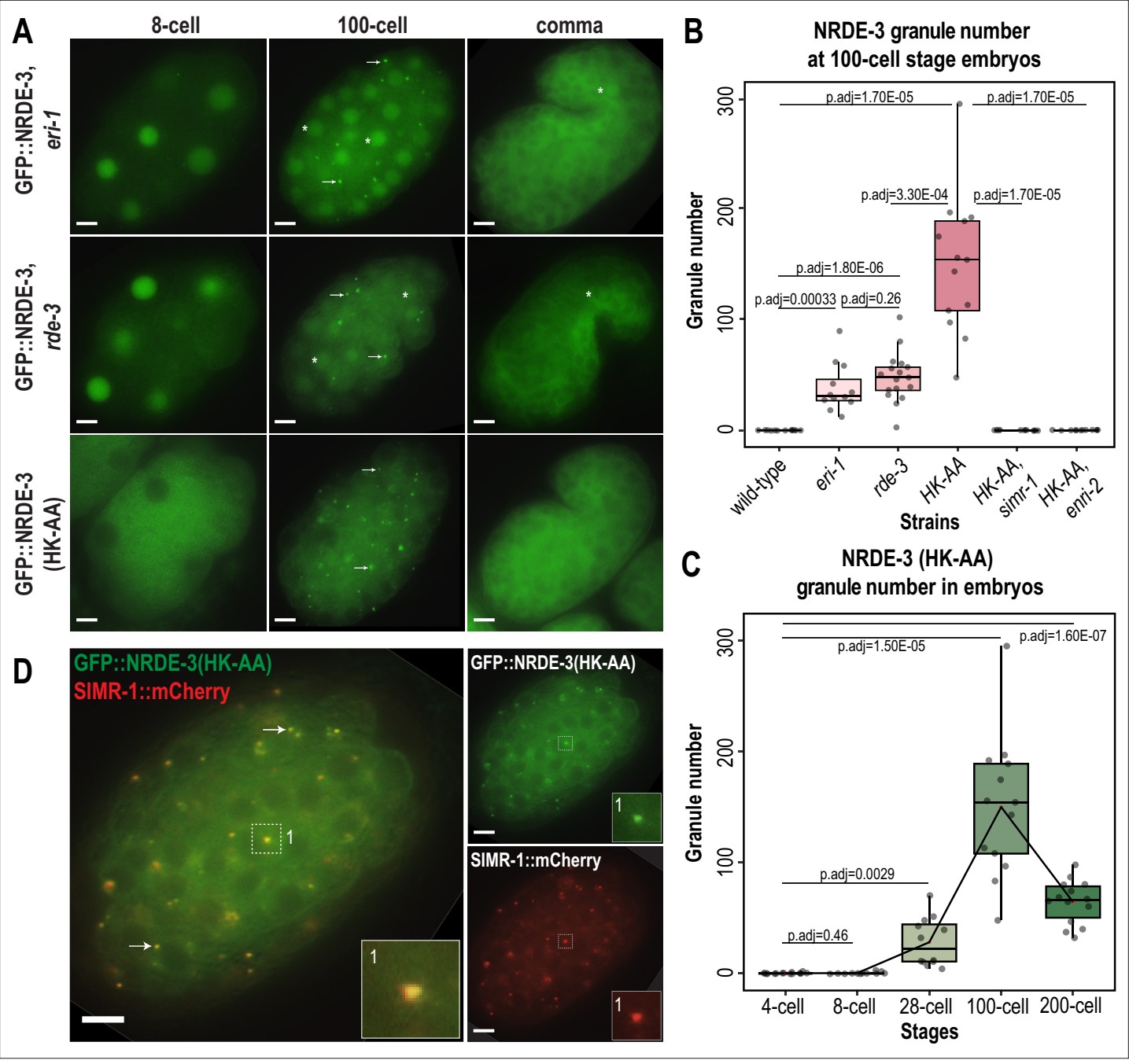

**Figure 2.** Unloaded NRDE-3 localizes to cytoplasmic granules with SIMR-1. (**A**) Live imaging of GFP::3xFLAG::NRDE-3 embryos in *eri-1*, *rde-3*, and *nrde-3(HK-AA)* mutants at 8 cell, 100 cell, and comma stage embryos. At least five individual embryos were imaged for each genotype and stage. Arrows point to granule localization of NRDE-3 in the 100 cell stage. Asterisks highlight the nuclear localization of NRDE-3 in somatic cells of the 100 cell stage embryos and in the primordial germ cells of the comma stage embryos. Scale bars, 5 µm. (**B**) Box plot of GFP::3xFLAG::NRDE-3 granule number quantification in different mutants. (**C**) Box plot of GFP::3xFLAG::NRDE-3(HK-AA) granule number quantification at different embryonic stages. Lines connect the mean granule number (red dots) for each stage, illustrating the change in number of NRDE-3 granules across embryonic development. (**D**) Live imaging of SIMR-1::mCherry::2xHA; GFP::3xFLAG::NRDE-3(HK-AA) at 100 cell stage. Arrows and insets show examples of colocalization between SIMR-1 and NRDE-3(HK-AA). At least ten individual embryos were imaged. Scale bars, 5 µm. For box plots in B and C, at least twelve individual embryos in each mutant were used for quantification. Each dot represents an individual embryo, and all data points are shown. Bolded midline indicates median value, box indicates the first and third quartiles, and whiskers represent the most extreme data points within 1.5 times the interquartile range. Two-tailed t-tests were performed to determine statistical significance and p-values were adjusted for multiple comparisons. See Materials and methods for a detailed description of quantification methods.

*Figure 2 continued on next page*

*Figure 2 continued*

The online version of this article includes the following source data and figure supplement(s) for figure 2:

**Source data 1.** This file contains the raw data used to generate the graphs of NRDE-3 granule number shown in *Figure 2B and C*.

**Figure supplement 1.** NRDE-3 and SIMR-1 localization and expression in various mutants.

**Figure supplement 1—source data 1.** This file contains the original western blots of GFP::NRDE-3 used in creating *Figure 2—figure supplement 1B*, indicating the relevant bands, antibodies, strains, and developmental stages.

**Figure supplement 1—source data 2.** This file contains the original files for western blot analysis displayed in *Figure 2—figure supplement 1B*.

**Figure supplement 1—source data 3.** This file contains the raw data used to generate the graphs of NRDE-3 granule number shown in *Figure 2—figure supplement 1D*.

be a fundamental difference in the small RNAs bound by NRDE-3 in early compared to late embryos, and that the small RNAs bound by NRDE-3 in early embryos are produced independently of ERI-1 and RDE-3.

To determine whether unloaded NRDE-3 localizes to SIMR granules, we examined the localization of SIMR-1 and NRDE-3 together in the *nrde-3(HK-AA)* mutant and *eri-1* mutant backgrounds. We found that SIMR-1 colocalizes with unloaded NRDE-3 in embryonic granules (*Figure 2D*, *Figure 2—figure supplement 1C*). Further, the SIMR granules in the *nrde-3(HK-AA)* mutant background exhibit dynamics similar to the wild-type background (*Figure 2—figure supplement 1D*), indicating that *nrde-3(HK-AA)* does not affect the localization of SIMR-1. Interestingly, NRDE-3(HK-AA) does not form granules in germ cells and is instead present exclusively in the cytoplasm, thus it does not colocalize with the SIMR compartment of germ granules (*Figure 2—figure supplement 1E*). These results demonstrate that unloaded NRDE-3 associates with SIMR-1 and ENRI-2 in cytoplasmic granules in the somatic cells of *C. elegans* embryos, indicating a potential role for SIMR-1 in the NRDE-3 nuclear RNAi pathway.

## SIMR-1 and ENRI-2 recruit unloaded NRDE-3 to cytoplasmic granules

As previously described, unloaded NRDE-3 localizes to cytoplasmic granules in embryos and colocalizes with SIMR-1. Next, we aimed to determine whether SIMR-1 and ENRI-2 are required for the NRDE-3 granule localization. To this end, we introduced a *simr-1* mutant and an *enri-2* mutant into the GFP-tagged NRDE-3(HK-AA) strain and assessed NRDE-3(HK-AA) localization. Strikingly, we found that NRDE-3(HK-AA) granules disappear completely and NRDE-3(HK-AA) is instead found exclusively in the cytoplasm in all cells across all embryonic stages (*Figure 3A*). Similarly, in a *simr-1; eri-1* double mutant, NRDE-3 granules are absent though NRDE-3 remains in the nucleus in early embryos (*Figure 3A*), similar to NRDE-3 expression in the *eri-1* single mutant (*Figure 2A*). However, neither wild-type NRDE-3 nor NRDE-3(HK-AA) protein levels are affected by loss of *simr-1* (*Figure 3—figure supplement 1A*), indicating that loss of granule localization in *nrde-3(HK-AA); simr-1* is not due to reduced NRDE-3 protein expression. These results demonstrate that both SIMR-1 and ENRI-2 are required for the recruitment of NRDE-3 to cytoplasmic granules.

To investigate the dependence of SIMR-1 and ENRI-2 on one another, we examined ENRI-2 localization in a *simr-1* mutant and SIMR-1 localization in an *enri-2* mutant. We found that ENRI-2 granules are lost in the *simr-1* mutant, while SIMR-1 granules are still present in the *enri-2* mutant, indicating that SIMR-1 functions upstream of ENRI-2 for granule assembly (*Figure 3B*).

To further assess whether ENRI-1 plays a role in the accumulation of unloaded NRDE-3 in cytoplasmic granules, we introduced an *enri-1* mutant into the GFP-tagged NRDE-3(HK-AA) strain and found that NRDE-3 association with cytoplasmic granules was not disrupted (*Figure 3—figure supplement 1B*). We further examined NRDE-3(HK-AA) granule association in the *enri-1; enri-2* double mutant and found it to be fully cytoplasmic, indistinguishable from the *enri-2* single mutant (*Figure 3—figure supplement 1B*). While we had already determined that ENRI-1::mCherry did not form visible foci in embryos; to rule out the possibility of partial redundancy between ENRI-1 and ENRI-2, we introduced an *enri-2* mutant into the mCherry-tagged ENRI-1 strain but still unable to detect any distinct ENRI-1 expression (*Figure 3—figure supplement 1C*). Lastly, to determine if NRDE-3 recruitment to granules could alter ENRI-1 localization, we introduced the mCherry-tagged ENRI-1 into the GFP-tagged NRDE-3(HK-AA) strain, and still we could not see any granule localization for ENRI-1 (*Figure 3—figure supplement 1D*). Therefore, we excluded ENRI-1 from further investigation, and conclude that

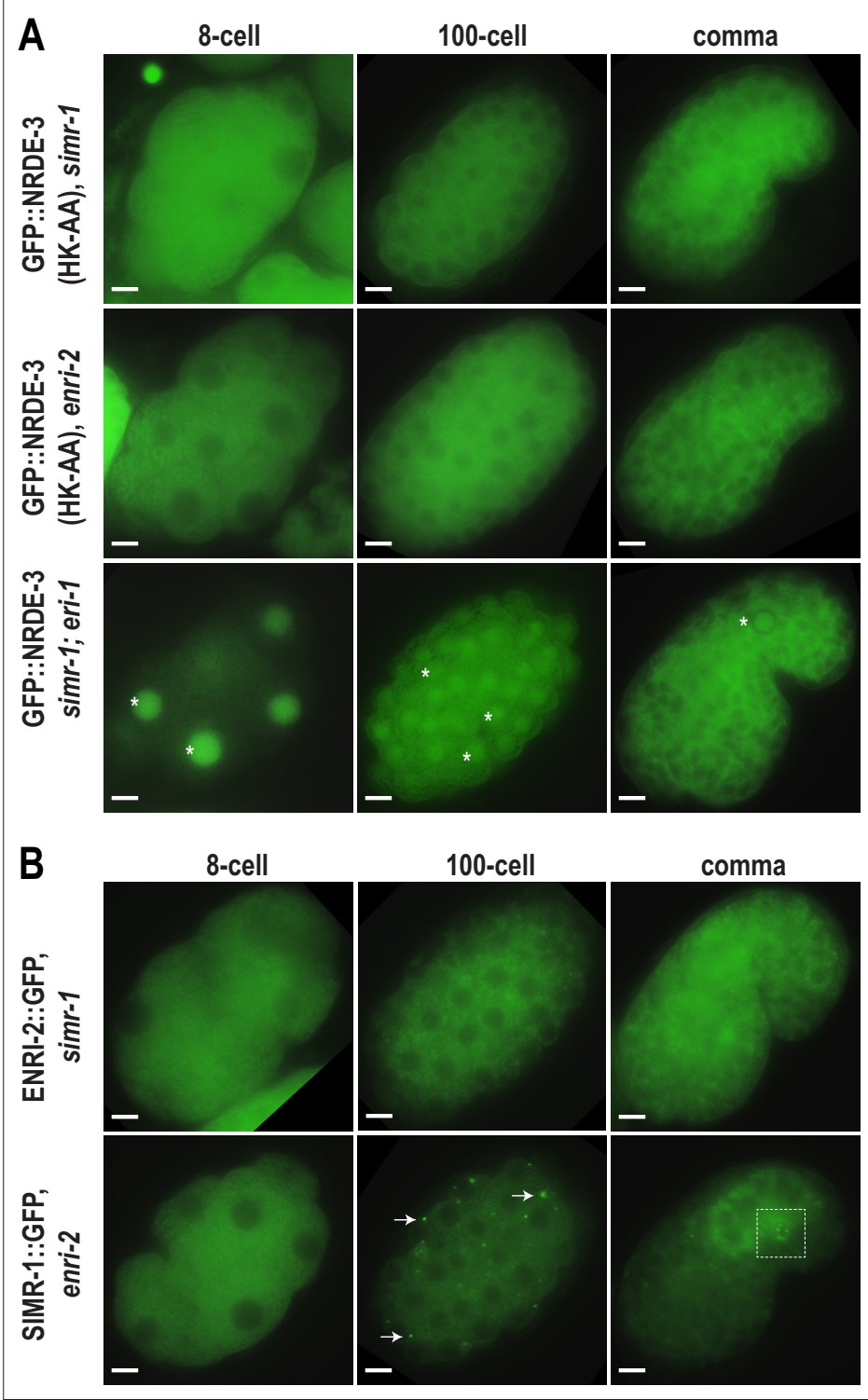

**Figure 3.** SIMR-1 recruits ENRI-2 and then NRDE-3 to cytoplasmic granules. (**A**) Live imaging of GFP::3xFLAG::NRDE-3(HK-AA) embryos in *simr-1* and *enri-2* mutants, and GFP::3xFLAG::NRDE-3 embryos in a *simr-1; eri-1* double mutant at 8 cell, 100 cell, and comma stages. At least five individual embryos were imaged for each genotype and stage. Asterisk marks the nuclear localization of GFP::NRDE-3 in the *simr-1; eri-1* mutant, visible in somatic cells of 8- and 100- cell stage embryos and in a primordial germ cell of the comma stage embryo. Scale bars, 5 μm. (**B**) Live imaging of ENRI-2::2xTy1::GFP embryos in a *simr-1* mutant and SIMR-1::GFP::3xFLAG embryos in an *enri-2* mutant. At least five individual embryos were imaged for each genotype and stage. Arrows

*Figure 3 continued on next page*

*Figure 3 continued*

point to examples of cytoplasmic SIMR granules still visible in the *enri-2* mutant. Box surrounds a primordial germ cell displaying germ granule localization of SIMR-1. Scale bars, 5 μm.

The online version of this article includes the following source data and figure supplement(s) for figure 3:

**Figure supplement 1.** NRDE-3(HK-AA) and ENRI-1 localization in various mutants.

**Figure supplement 1—source data 1.** This file contains the original western blots of GFP::NRDE-3 used in creating *Figure 3—figure supplement 1A*, indicating the relevant bands, strains, and antibodies.

**Figure supplement 1—source data 2.** This file contains the original files for western blot analysis displayed in *Figure 3—figure supplement 1A*.

SIMR-1 and ENRI-2, but not ENRI-1, recruit unloaded NRDE-3 to cytoplasmic granules, with SIMR-1 also acting to recruit ENRI-2.

## SIMR-1 does not localize to P bodies or other previously identified embryonic granules

A variety of RNA-associated proteins have previously been shown to form granules in *C. elegans* embryos. To determine whether the SIMR-1, ENRI-2, and unloaded NRDE-3 granules that we have observed coincide with a previously identified granule, we examined the colocalization between SIMR-1 and all other embryonic granule-associated proteins that we could identify. It is well known that Processing (P) bodies, the condensates of translationally inactive mRNAs and proteins, localize to cytoplasmic foci of soma in *C. elegans* embryos (*Parker and Sheth, 2007*; *Gallo et al., 2008*). To examine if the SIMR-1 cytoplasmic granules are P bodies, we examined the localization of SIMR-1 and CGH-1, a core P body component, using a strain expressing GFP-tagged SIMR-1 and mCherry-tagged CGH-1 (*Du et al., 2023*). We found that CGH-1 does not colocalize with SIMR-1 (*Figure 4—figure supplement 1A*). CGH-1 also does not colocalize with NRDE-3 cytoplasmic granules in the *eri-1* mutant (*Figure 4—figure supplement 1B*). Together, these data indicate that the cytoplasmic SIMR granules found in embryos are not P bodies.

Next, we examined two proteins previously shown to colocalize with SIMR foci in the germ cells of adult animals, RSD-2 and HRDE-2 (*Manage et al., 2020*; *Chen and Phillips, 2024a*). RSD-2 is a small RNA factor required for the response to low doses of exogenously-introduced double-stranded RNA (*Sakaguchi et al., 2014*; *Han et al., 2008*; *Tijsterman et al., 2004*; *Zhang et al., 2012*) and HRDE-2 is a factor critical for RNAi inheritance that promotes correct small RNA loading into the nuclear Argonaute HRDE-1 (*Chen and Phillips, 2024a*; *Spracklin et al., 2017*). However, we did not observe any granule localization for RSD-2 and HRDE-2 in embryos (*Figure 4—figure supplement 1C–D*). In addition, SIMR-1 cytoplasmic granules were not affected by the loss of *hrde-2* (*Figure 4—figure supplement 1E*). These results suggest that HRDE-2 and RSD-2 do not function together with SIMR-1, ENRI-2, and NRDE-3 in embryonic granules and indicate that embryonic SIMR granules and the SIMR compartment of germ granules are distinct, each containing a unique set of proteins.

RDE-12 interacts with Argonaute proteins and RNAi-targeted mRNAs, and has also been shown to localize to cytoplasmic granules in the somatic cells of *C. elegans* embryos (*Shirayama et al., 2014*; *Yang et al., 2014*). We next assessed the localization of mCherry-tagged SIMR-1 relative to GFP-tagged RDE-12 and found that they do not colocalize (*Figure 4—figure supplement 1F*). RSD-6 is a Tudor domain-containing RNAi factor that partially colocalizes with RDE-12 in the R2 bodies in adult germ cells (*Yang et al., 2014*; *Sakaguchi et al., 2014*; *Zhang et al., 2012*). We examined the expression of GFP-tagged RSD-6 in embryos and did observe RSD-6 at granules in somatic cells, while no colocalization with mCherry-tagged SIMR-1 could be detected (*Figure 4—figure supplement 1G*). The RNAi-inheritance factor and defining member of the Z compartment of the germ granule, ZNFX-1, has also been observed in cytoplasmic granules in the somatic cells of embryos (*Wan et al., 2018*; *Ouyang et al., 2019*), however these somatic ZNFX-1 granules also fail to colocalize with SIMR-1 (*Figure 4—figure supplement 1H*). Finally, we compared SIMR-1 localization to peri-centrosomal foci marked tubulin, based on a recent report that NRDE-3 accumulates at these foci (*Jin et al., 2024*), but we did not observe any colocalization between SIMR-1 and tubulin in a wild-type background (*Figure 4—figure supplement 1I*). Altogether, we found that SIMR-1 fails to localize to any previously characterized embryonic granules. These results further indicate that there

are numerous granule-localized proteins in the somatic cells of embryos, such as RDE-12, RSD-6, and ZNFX-1, which may play important roles in the RNA biology of early embryos.

## Multiple *Mutator* complex proteins localize to SIMR granules in embryos

*Mutator* foci localize adjacent to SIMR foci in the adult germline (*Manage et al., 2020*; *Chen and Phillips, 2024a*), so we next investigated the localization of *Mutator* components in embryos. We first examined *Mutator* foci component RDE-3/MUT-2, a poly(UG) polymerase required for WAGO-class 22G-RNA production (*Figure 1—figure supplement 1A–B*; *Phillips et al., 2012*; *Shukla et al., 2020*) and found that GFP-tagged RDE-3 is prominently localized to cytoplasmic granules in embryos that colocalize with SIMR-1 (*Figure 4A–B*). This colocalization led us to the hypothesis that SIMR-1 cytoplasmic granules are sites of WAGO-class 22G-RNA biogenesis. Therefore, we speculated that more small RNA production machinery might be localized with SIMR-1 at these cytoplasmic granules. We next examined the RNA-dependent-RNA-polymerase (RdRP) RRF-1, which synthesizes WAGO-class 22G-RNAs and localizes to *Mutator* foci in the adult germline (*Figure 1—figure supplement 1A–B*; *Sijen et al., 2001*; *Gent et al., 2010*; *Vasale et al., 2010*; *Phillips et al., 2012*). As we predicted, RRF-1 also colocalizes with SIMR-1 in somatic granules (*Figure 4B–C*), and it fails to localize to somatic granules in the *simr-1* mutant (*Figure 4D*).

MUT-16 is the scaffolding protein for germline *Mutator* foci, thus we next investigated whether MUT-16 similarly scaffolds the cytoplasmic SIMR granules in early embryos (*Phillips et al., 2012*). We found that MUT-16 can be observed in cytoplasmic granules in the embryonic somatic cells (*Figure 4—figure supplement 1J*), similar to what has been observed in a previous study (*Ouyang et al., 2019*), and both SIMR-1 and RDE-3 fail to assemble into cytoplasmic granules in the *mut-16* mutant (*Figure 4E*, *Figure 4—figure supplement 1K*). Notably, the germ granule association of SIMR-1 is unaffected, as SIMR-1 still localizes to germ granules at comma stage embryos and in the adult germline (*Figure 4E*; *Manage et al., 2020*). Together, these data indicate that MUT-16 functions upstream of SIMR-1 and mediates the assembly of cytoplasmic granules in embryos. It is curious to note that, in a *mut-16* mutant where SIMR-1 association with cytoplasmic granules is lost in the somatic cells, SIMR-1 instead associates with mitotic spindles (*Figure 4E*), similar to the localization observed for NRDE-3 in embryos (*Jin et al., 2024*). To conclude, we have shown that the SIMR granules found in the somatic cells of early embryos contain the biogenesis machinery for WAGO-class 22G-RNAs, including RDE-3 and RRF-1, and depend on the scaffolding protein MUT-16 for assembly (*Figure 4J*). The differential requirement for MUT-16 on the assembly of somatic and germline SIMR granules highlights a key difference between these two compartments, which have some parallel functions but distinct composition.

## CSR-1 and EGO-1 associate with a distinct type of granule in early embryos

The Argonaute protein CSR-1 has also been previously seen at cytoplasmic granules in the soma of early embryos, a time at which CSR-1 is functioning to clear maternal-inherited mRNAs (*Quarato et al., 2021*; *Seroussi et al., 2023*; *Ouyang et al., 2019*). Using a GFP-tagged CSR-1 strain we constructed previously (*Nguyen and Phillips, 2021*), we confirmed that CSR-1 forms prominent cytoplasmic granules in embryos visible prior to the 4 cell stage and present through 100 cell stage embryos, but disappear by the 200 cell stage, at which point only germ granule localization is visible (*Figure 4F*). Quantification of the total number of 'CSR granules' per embryo across development shows that the CSR granules are more abundant than the SIMR-1 and NRDE-3(HK-AA) granules and differ in the timing of their appearance and disappearance relative to SIMR-1 and NRDE-3(HK-AA) granules (*Figures 1C, 2C and 4G*). Specifically, CSR granules appear earlier and peak at the 28 cell stage, while SIMR-1 and NRDE-3(HK-AA) granules appear between 8- and 28 cell stages and peak at the 100 cell stage (*Figures 1C, 2C and 4G*). The small RNAs bound by CSR-1 are synthesized by the RdRP, EGO-1, so we next assessed the localization of EGO-1 in early embryos. We found that EGO-1 colocalizes with CSR-1 in the somatic CSR granules (*Figure 4B and H*) and neither CSR-1 nor EGO-1 fully colocalizes with SIMR-1, although we occasionally observed adjacent localization between SIMR-1 and CSR-1 (*Figure 4B and I*, *Figure 4—figure supplement 1L*). Lastly, unlike RRF-1 which requires SIMR-1 to localize to embryonic foci, EGO-1 localizes to cytoplasmic granules in the absence

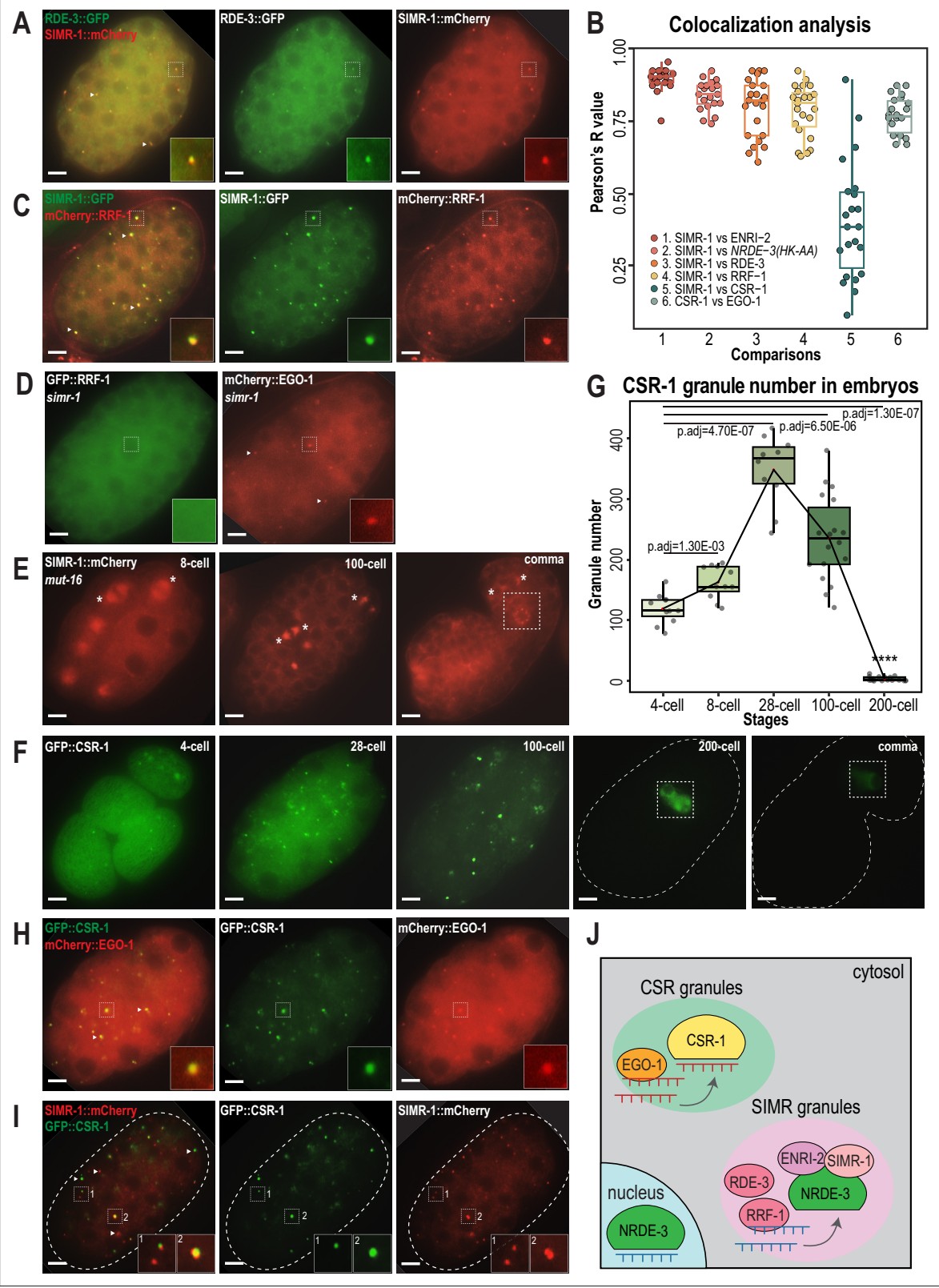

**Figure 4.** CSR and WAGO pathway proteins localize to distinct cytoplasmic granules. (**A**) Live imaging of SIMR-1::mCherry::2xHA; RDE-3::GFP embryo at 100 cell stage, showing that RDE-3 colocalizes with SIMR-1. At least five individual embryos were imaged for each genotype and stage. Arrowheads and insets show examples of colocalization between SIMR-1 and RDE-3 at cytoplasmic granules. Scale bars, 5 μm. (**B**) Box plot of Pearson's R value quantifications among different pairs of proteins at 100 cell embryonic stage. At least 20 granules from at least four individual embryos were used for

*Figure 4 continued on next page*

*Figure 4 continued*

quantification. Each dot represents an individual quantification, and all data points are shown. Box indicates the first and third quartiles, and whiskers represent the most extreme data points within 1.5 times the interquartile range. See Materials and methods for a detailed description of quantification methods. (**C**) Live imaging of SIMR-1::GFP::3xFLAG; HA::EGO-1::mCherry::RRF-1 at 100 cell stage embryo, showing that RRF-1 colocalizes with SIMR-1. At least five individual embryos were imaged for each genotype and stage. Arrowheads and insets show examples of colocalization between SIMR-1 and RRF-1 at cytoplasmic granules. Scale bars, 5 µm. (**D**) Live imaging of mCherry::EGO-1::GFP::RRF-1 in a *simr-1* mutant, showing that RRF-1 no longer associates with cytoplasmic granules, while EGO-1 remains associated with granules in the *simr-1* mutant. At least five individual embryos were imaged. Arrowheads point to examples of cytoplasmic EGO-1 granules in a *simr-1* mutant. Insets show examples of cytoplasmic localization of RRF-1 and granule localization of EGO-1 in a *simr-1* mutant. Scale bars, 5 µm. (**E**) Live imaging of SIMR-1::mCherry::2xHA embryos in a *mut-16* mutant at 8 cell, 100 cell, and comma stages. At least five individual embryos were imaged. Asterisks indicate spindle localization of SIMR-1 in a *mut-16* mutant. Box highlights germ granule localization of SIMR-1 in a comma-stage, *mut-16* mutant embryo. Scale bars, 5 µm. (**F**) Live imaging of GFP::3xFLAG::CSR-1 embryos at different stages (4 cell, 28 cell, 100 cell, 200 cell, and comma), shows that CSR-1 localizes to cytoplasmic granules in early embryos and is restricted to germ granules in late embryos. At least three individual embryos were imaged for each stage. Dotted white line marks perimeter of the embryo. Box marks germ granule localization of CSR-1. Scale bars, 5 µm. (**G**) Box plot quantifying GFP::3xFLAG::CSR-1 granules at different embryonic stages. At least ten embryos at each stage were used for quantification. Each dot represents an individual embryo, and all data points are shown. Bolded midline indicates median value, box indicates the first and third quartiles, and whiskers represent the most extreme data points within 1.5 times the interquartile range. Lines connect the mean granule number (red dots) at each stage, illustrating the change in number of CSR granules across embryonic development. Two-tailed t-tests were performed to determine statistical significance and p-values were adjusted for multiple comparisons. See Materials and methods for a detailed description of quantification methods. (**H**) Live imaging of mCherry::EGO-1; GFP::3xFLAG::::CSR-1 embryo at 28 cell stage, showing CSR-1 colocalization with EGO-1. At least 10 individual embryos were imaged. Arrowheads and insets show examples of CSR-1 and EGO-1 colocalization at cytoplasmic granules. Scale bars, 5 µm. (**I**) Live imaging of SIMR-1::mCherry::2xHA; GFP::3xFLAG::CSR-1 embryo at 28 cell stage, showing the absence of colocalization between SIMR-1 and CSR-1 with occasional adjacent localization. At least ten individual embryos were imaged. Arrowheads point to examples of SIMR and CSR granules that do not colocalize. Insets show examples of SIMR and CSR granules that are found adjacent to each other or fail to colocalize. Dotted white line marks perimeter of embryo. Scale bars, 5 µm. (**J**) Model of CSR and SIMR granules in the somatic cells of *C. elegans* embryos. The RdRP EGO-1, which synthesizes CSR-class 22G-RNAs, localizes to CSR granules, where CSR-1 loading may take place. The RdRP RRF-1, along with RDE-3, ENRI-2, and unloaded NRDE-3 localize to SIMR granules. SIMR-1 and ENRI-2 recruits unloaded NRDE-3 to granule where RRF-1 may synthesize ERGO-dependent 22G-RNAs for loading into NRDE-3. After loading, NRDE-3 translocates to the nucleus and silences genes co-transcriptionally.

The online version of this article includes the following source data and figure supplement(s) for figure 4:

**Source data 1.** This file contains the raw data used to generate the graphs of colocalization analysis and CSR-1 granule number shown in *Figure 4B and G*.

**Figure supplement 1.** SIMR-1 does not colocalize with any previously identified embryonic granules.

of SIMR-1 (*Figure 4D*). Together, our results show that the RdRPs, RRF-1 and EGO-1, localize to different cytoplasmic granules in the somatic cells of *C. elegans* embryos, where they colocalize with Argonaute proteins, NRDE-3 and CSR-1, respectively. Thus, we postulate that WAGO-class and CSR-class 22G-RNA biogenesis and loading are compartmentalized into cytoplasmic granules, differing from one another both spatially and temporally, in the somatic cells of early embryos (*Figure 4J*).

## Autophagy regulates the removal of SIMR granules and other cytoplasmic granules during embryogenesis

Both SIMR and CSR granules exhibit dynamic expression patterns across embryonic development, where the granules increase in abundance from early to mid-embryogenesis, and then subsequently disappear from somatic cells, becoming restricted to germ cells by late embryogenesis (*Figures 1B–C and 4F–G*). In contrast, P granules become enriched at the posterior half of one-cell embryos and are subsequently partitioned with the germline progenitor cells through the remainder of embryonic development (*Strome and Wood, 1983*). Previous work has shown that autophagic degradation is one mechanism by which P granules are removed from somatic blastomeres (*Zhang et al., 2009*). Autophagy, often referred to as the 'self-eating' pathway, is one of the two major protein degradation systems in eukaryotic cells. Autophagy involves the formation of autophagosomes, which engulf cytoplasmic structures and proteins and deliver the contents to lysosomes for degradation (*Klionsky, 2005*; *Mizushima, 2007*). In contrast, the ubiquitin-proteasome system degrades proteins though ubiquitination and degradation by the 26 S proteosome complex, bypassing lysosomes (*Nedelsky et al., 2008*). In the absence of the autophagy protein LGG-1, the *C. elegans* ortholog of *S. cerevisiae* Atg8, the core P granule proteins PGL-1 and PGL-3 are ectopically expressed in somatic foci during mid to late embryogenesis (*Zhang et al., 2009*). So we sought to test whether the removal of SIMR

and CSR granules from the somatic cells during mid-to-late embryogenesis might similarly depend on the autophagy pathway.

To address this possibility, we first examined the expression of PGL-1 and SIMR-1 at 8 cell, 100 cell, and comma stages of embryogenesis following *lgg-1* or control (L4440) RNAi treatment of the parental animals. Consistent with previous work (*Zhang et al., 2009*), RNAi of *lgg-1* leads to ectopic expression of PGL-1 in cytoplasmic foci of the somatic cells at mid (100 cell) and late (comma) embryonic stages, while its germ granule localization remains unaffected (*Figure 5A*). Similarly, we found that the number of somatic SIMR granules at the 100 cell stage increases by approximately 2-fold in *lgg-1* RNAi-treated embryos compared to control (L4440) (*Figure 5A and B*). Further, in the comma stage, somatic SIMR granules remain in the soma in *lgg-1* RNAi-treated embryos, whereas SIMR granules are cleared from the somatic cells and restricted to germ cells in control (L4440) RNAi-treated embryos (*Figure 5A*). Examination of other somatic SIMR granule-associated proteins, including ENRI-2, RDE-3, and RRF-1, and unloaded NRDE-3 revealed that they also exhibit ectopic expression in the somatic cells of comma-stage embryos and continue to colocalize with SIMR-1 (*Figure 5—figure supplement 1A and B*). These findings indicate that the removal of SIMR granules from somatic cells during mid and late embryogenesis is regulated by autophagy, similar to the role of autophagy in the degradation of P granules form early somatic blastomeres.

Since both SIMR granules and P granules are regulated by the autophagy pathway during embryogenesis, we also examined the role for autophagy in regulating other granule-associated proteins in early embryos, specifically CSR-1 and ZNFX-1. Similar to PGL-1 and SIMR-1, CSR-1 and ZNFX-1 show increased numbers of somatic granules in 100 cell and comma stage embryos following parental treatment with *lgg-1* RNAi compared to control (L4440) (*Figure 5C*). To further investigate the spatial organization of these somatic granules in the absence of autophagy-mediated degradation, we examined the 100 cell stage embryos following *lgg-1* RNAi treatment using strains expressing PGL-1::BFP; SIMR-1::GFP; RFP::ZNFX-1, and GFP::CSR-1; SIMR-1::mCherry. Interestingly, these proteins display a variety of configurations relative to one another, distinct from their typical organization in the germline (*Uebel et al., 2023*). These configurations included granules composed of individual protein, multiple proteins adjacent to one another, and large aggregates composed of multiple proteins (*Figure 5D*). We speculate that many of the PGL-1-containing granules, including the larger aggregates, are associated with autophagosomes, as has been shown previously (*Zhang et al., 2009*). Overall, our findings demonstrate that the autophagy pathway is utilized to regulate the spatial-temporal expression of many embryonic granule-associated proteins, including proteins found in P, Z, SIMR, and CSR granules. Further, it is worth noting that P, Z, SIMR, and CSR granules are not all removed at the same developmental timepoint (for example, *Figures 1C and 4G*), suggesting the presence of additional regulatory mechanisms controlling the timing of their degradation.

## NRDE-3 switches small RNA partners during embryonic development

The nuclear localization of NRDE-3 in the somatic cells of larvae depends on ERGO-1 and other proteins required for the biogenesis of ERGO-class 26G-RNAs (*Figure 1—figure supplement 1A–B*; *Guang et al., 2008*). Sequencing of NRDE-3-bound 22G-RNAs at the L4 to young adult transition identified a set of endogenous targets that overlap substantially with those of ERGO-1 (*Seroussi et al., 2023*). Together, these data have led to the conclusion that NRDE-3 acts downstream of ERGO-1 to transcriptionally silence ERGO-target genes. Yet our data looking at the nuclear localization of NRDE-3 in embryos, demonstrate that this model may be an incomplete picture. Specifically, in *eri-1* and *rde-3* mutants where 26G-RNA or WAGO-class 22G-RNA biogenesis are abolished, respectively, NRDE-3 remains localized to the nucleus in early embryos (*Figure 2A*). The small RNA binding-defective NRDE-3(HK-AA) is localized exclusively to the cytoplasm at the same time point, indicating that small RNA binding is critical for nuclear import at this stage (*Figure 2A*). Accordingly, we must postulate that NRDE-3 binds another class of small RNA to promote nuclear entry in very early embryos. To investigate the identity of NRDE-3-bound small RNAs across embryonic development and to explore the role of the SIMR granules in promoting NRDE-3 small RNA binding, we immunoprecipitated NRDE-3 and sequenced associated small RNAs (IP-sRNA seq) in early embryos (≤ 100 cell) and late embryos (≥ 300 cell) in wild-type, *eri-1* mutant, *simr-1* mutant, and *enri-2* mutant animals (*Figure 6A*).

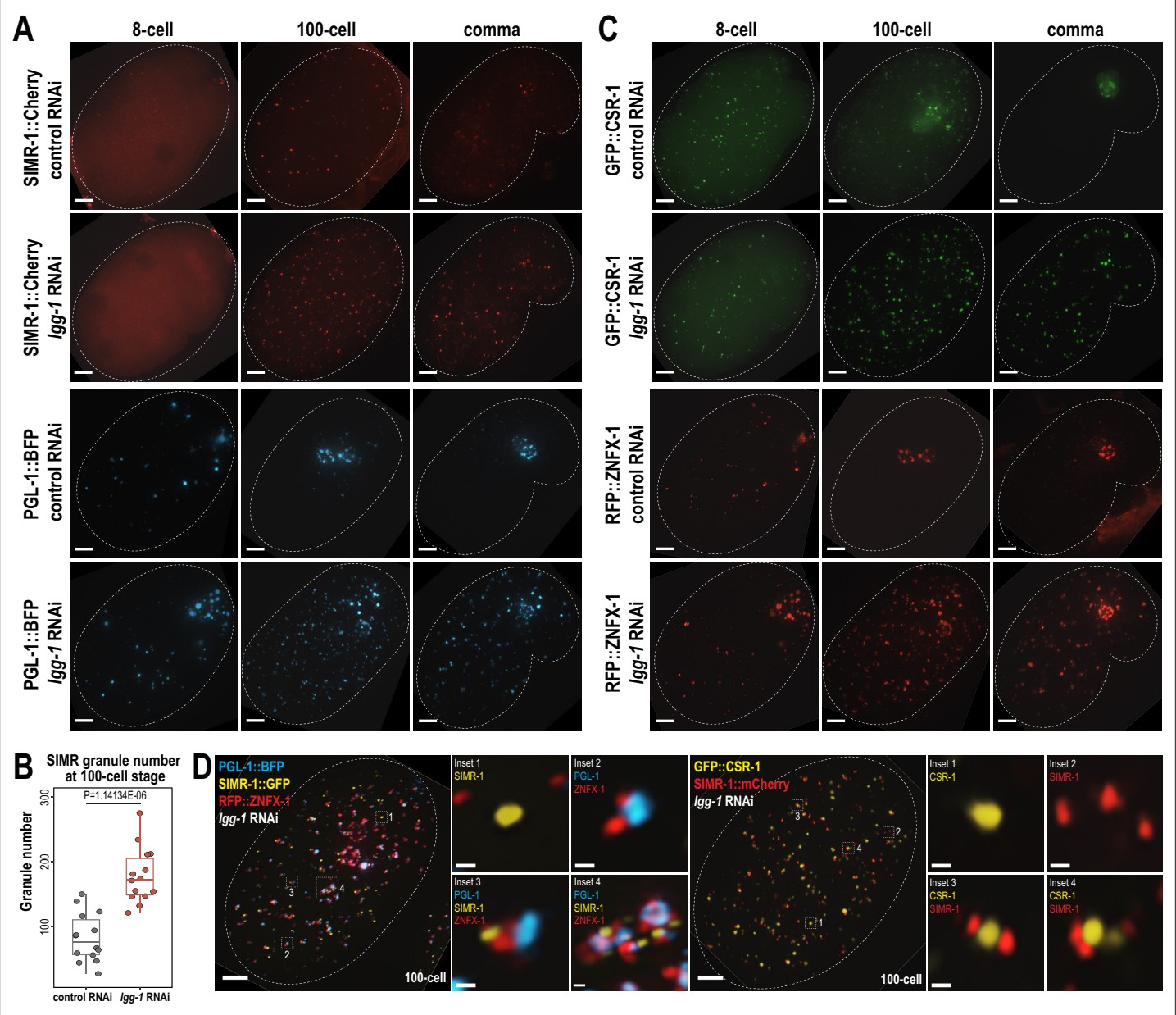

**Figure 5.** Autophagy regulates the removal of SIMR granules and other embryonic granules. (**A**) Live imaging of SIMR-1::mCherry and PGL-1::BFP at 8 cell, 100 cell, and comma stage embryos following treatment of parental animals with control (L4440) and *lgg-1* RNAi. At least ten individual embryos were imaged for each genotype and stage. Dotted white line marks perimeter of embryo. Scale bars, 5 µm. (**B**) Box plot quantifying the number of SIMR granules at the 100 cell stage following treatment of parental animals with control (L4440) and *lgg-1* RNAi. Fourteen embryos at each stage were used for quantification. Each dot represents an individual embryo, and all data points are shown. Bolded midline indicates median value, box indicates the first and third quartiles, and whiskers represent the most extreme data points within 1.5 times the interquartile range. Two-tailed t-test was performed to determine statistical significance. (**C**) Live imaging of GFP::CSR-1 and RFP::ZNFX-1 at 8 cell, 100 cell, and comma stage embryos following treatment of parental animals with control (L4440) and *lgg-1* RNAi. At least ten individual embryos were imaged for each genotype and stage. Dotted white line marks perimeter of embryo. Scale bars, 5 µm. (**D**) Live imaging of SIMR-1::GFP; RFP:ZNFX-1; PGL-1::BFP embryo and SIMR-1::mCherry; GFP::CSR-1 embryo at 100 cell stage embryos following treatment of parental animals with *lgg-1* RNAi. At least ten individual embryos were imaged for each genotype. Dotted white line marks perimeter of embryo. Insets show examples of granule adjacency. Scale bars for embryos, 5 µm. Scale bars for insets, 0.2 µm. Images in A, C–D are maximum projections of deconvolved 12.5 µm z-stacks (about two-third of embryo depth).

The online version of this article includes the following source data and figure supplement(s) for figure 5:

**Source data 1.** This file contains the raw data used to generate the graph of SIMR granule number shown in *Figure 5B*.

**Figure supplement 1.** Components of the SIMR granule are regulated by autophagy.

**Figure supplement 1—source data 1.** This file contains the raw data used to generate the graphs of colocalization analysis shown in (B).

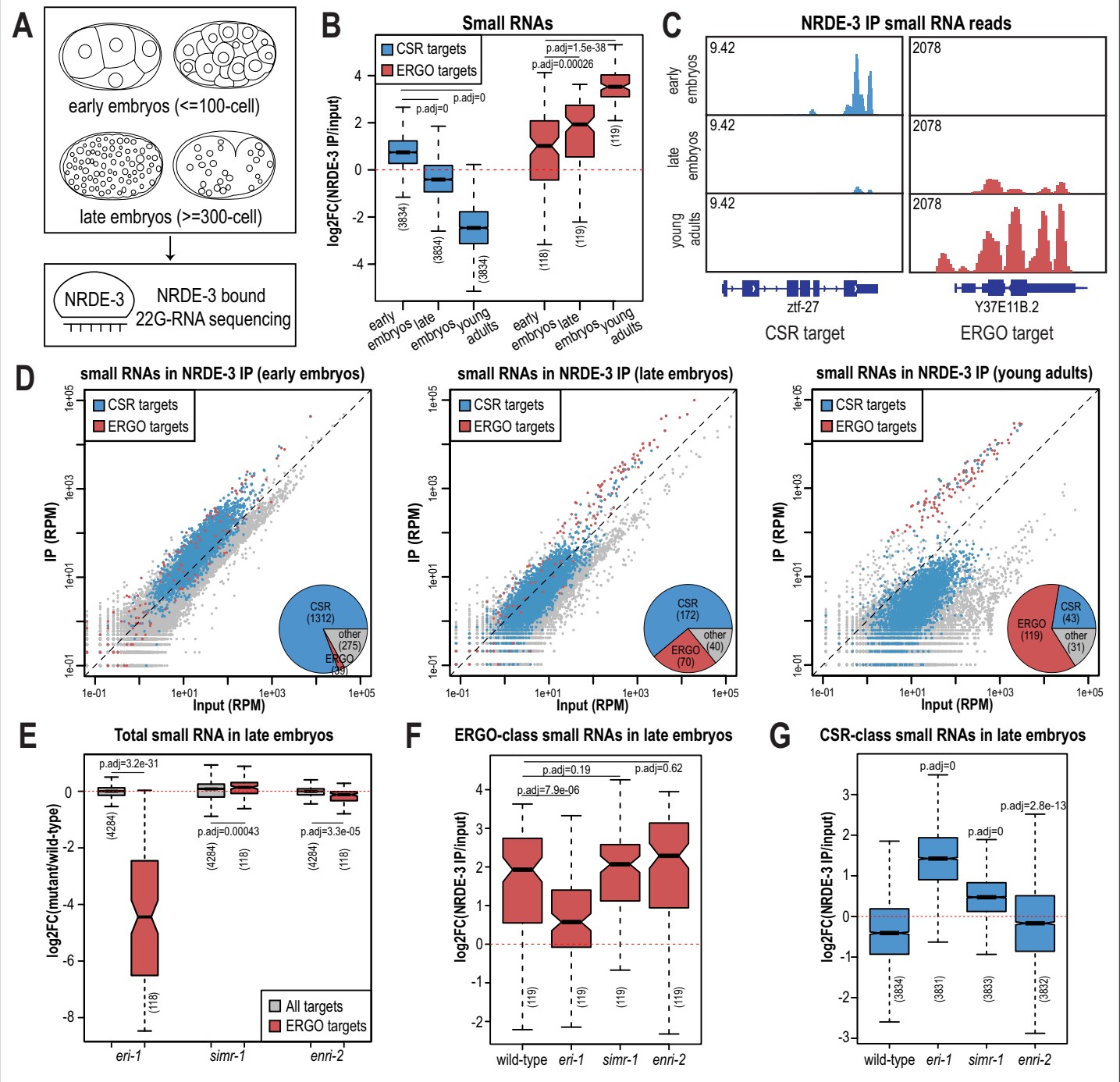

**Figure 6.** NRDE-3 switches small RNA targets during development. (**A**) Diagram of IP-sRNA seq on NRDE-3 early embryos (≤ 100 cell stage) and late embryos (≥ 300 cell). GFP::FLAG::NRDE-3 was immunoprecipitated from embryo lysate and its associated small RNAs were isolated for sequencing. (**B**) Box plot depicting log₂(fold change small RNA abundance) in NRDE-3 IP compared to input for at least two biological replicates. (**C**) Normalized NRDE-3-bound small RNA read distribution across a CSR-target gene (*ztf-27*) and an ERGO-target gene (Y37E11B.2) in early embryos, late embryos, and young adults. One representative replicate is shown. (**D**) Normalized NRDE-3 IP compared to input small RNA reads in early embryos, late embryos, and young adults. CSR-target and ERGO-target genes are indicated in blue and red, respectively. One representative replicate is shown. Insets are pie charts describing numbers of CSR targets, ERGO targets, and other targets that are significantly enriched in the NRDE-3 IP. The enriched targets were defined as small RNAs with at least 2-fold enrichment in IP compared to input, average RPM >10, and p-values ≤0.05. (**E**) Box plot depicting log₂(fold change small RNA abundance) in mutants compared to wild-type in late embryos for two or three biological replicates. (**F**) Box plot depicting log₂(fold change of ERGO-class small RNA abundance) in NRDE-3 IP compared to input in wild-type and mutants in late embryos for two or three biological replicates. (**G**) Box plot depicting log₂(fold change of CSR-class small RNA abundance) in NRDE-3 IP compared to input in wild-type and mutants in late

*Figure 6 continued on next page*

Figure 6 continued

embryos for two or three biological replicates. For box plots in B,E-G, bolded midline indicates median value, box indicates the first and third quartiles, and whiskers represent the most extreme data points within 1.5 times the interquartile range, excluding outliers. Two-tailed t-tests were performed to determine statistical significance and p-values were adjusted for multiple comparisons.

The online version of this article includes the following figure supplement(s) for figure 6:

**Figure supplement 1.** Defining NRDE-3-bound ERGO-target genes.

**Figure supplement 2.** NRDE-3-bound small RNA in different mutants.

Prior to analyzing our data, we sought to better define the expected NRDE-3-bound small RNAs. We initially planned to use two previously defined ERGO-target gene lists: the first list (ERGO - Manage) is defined by small RNAs significantly depleted at least two-fold in *ergo-1* mutant compared to wild-type at the gravid adult stage, with at least 10 reads per million (RPM) in wild-type samples and a DESeq2 adjusted p-value of <0.05 (*Manage et al., 2020*); the second list (ERGO - Fischer) is defined by genes reduced by 67% in *eri-7* adults or an average of 67% in *ergo-1, eri-1, eri-6*, and *eri-7* embryos, with at least 10 RPM in wild-type (*Fischer et al., 2011*). However, there is a poor overlap between these two datasets (*Figure 6—figure supplement 1D*), and small RNAs targeting many of these previously defined ERGO targets were not enriched by NRDE-3 in a published NRDE-3 IP-sRNA seq data on young adult animals that have begun oogenesis but do not yet have embryos (*Seroussi et al., 2023*; *Figure 6—figure supplement 1A–B*). To define a more stringent NRDE-3-target gene list at the young adult stage, we chose genes with at least four-fold enrichment (log$_2$FC ≥2) and 100 RPM (RPM ≥100) from the NRDE-3 IP-sRNA seq in young adults (*Seroussi et al., 2023*). This new list contains 119 genes and largely overlaps with the two previously defined ERGO-target gene lists (*Figure 6—figure supplement 1C–D*). To further confirm that this newly defined gene list represents NRDE-3 targets, we analyzed published small RNA and mRNA sequencing data from wild-type and *nrde-3* mutant mixed-stage embryos (before the bean stage) (*Padeken et al., 2021*). Compared to the Manage and Fischer ERGO-target gene lists, the NRDE-3-target gene list shows more significant small RNA depletion and a greater increase in mRNA expression in the *nrde-3* mutant compared to wild-type (*Figure 6—figure supplement 1A–C and E–F*). Therefore, we use the new NRDE-3-target gene list to represent the ERGO-1 pathway-dependent, NRDE-3-target genes (referred to here as ERGO targets) in the rest of this study.

We next examined the small RNAs bound to NRDE-3 in wild-type early embryos and late embryos, comparing our data to the published NRDE-3 IP-sRNA seq data on young adult animals (*Seroussi et al., 2023*). Strikingly, we found that in early embryos, the majority of small RNAs bound by NRDE-3 are CSR-class 22G-RNAs, which become progressively less enriched as the animals develop into late embryos and then young adults (*Figure 6B–D*). In contrast, enrichment for small RNAs targeting ERGO-target genes increases as *C. elegans* develops, and they become the majority of NRDE-3-bound small RNAs by young adulthood (*Figure 6B–D*). NRDE-3 also binds to CSR-target genes in the early embryos of the *eri-1* mutant, when it is observed to localize to the nucleus, indicating that the production of these NRDE-3-bound CSR-class 22G-RNAs is independent of *eri-1* and that CSR-class 22G-RNAs are likely sufficient to promote nuclear entry of NRDE-3 in the early embryo (*Figure 2A*, *Figure 6—figure supplement 2A*). To conclude, NRDE-3 binds to CSR-class 22G-RNA in early embryos but switches to bind preferentially to ERGO-dependent 22G-RNA in late embryos and young adults, suggesting that NRDE-3 may have two separable functions at distinct developmental time points. It is also curious to note that the change in small RNA preference of NRDE-3 coincides with the appearance and disappearance of the cytoplasmic SIMR granules, suggesting a potential role for SIMR-1 and ENRI-2 in promoting the switch of small RNA loading of NRDE-3.

## SIMR-1 and ENRI-2 contributes to an efficient switch of NRDE-3 bound small RNAs

Since ERGO-dependent 22G-RNA loading was mainly observed in late embryos, we focused on NRDE-3-bound small RNAs in the *eri-1* mutant, *simr-1* mutant, and *enri-2* mutant late embryos to determine the role of SIMR granules in promoting NRDE-3 small RNA binding specificity. We first examined the levels of ERGO-dependent small RNAs in the total small RNA samples and observed depletion of small RNAs mapping to ERGO-target genes in the *eri-1* mutant (*Figure 6E*,

*Figure 6—figure supplement 2A*). This result is consistent with previous research indicating that ERI-1 is required for ERGO-class 26G-RNA production and downstream ERGO-dependent 22G-RNA production (*Vasale et al., 2010*; *Han et al., 2009*; *Guang et al., 2008*). ERGO-dependent small RNAs are not substantially depleted in *simr-1* or *enri-2* mutants, indicating that RRF-1 can still synthesize a similar amount of ERGO-dependent 22G-RNAs when the cytoplasmic SIMR granules are absent (*Figure 6E*). Following NRDE-3 immunoprecipitation in the *eri-1* mutant, we observed a reduction in NRDE-3 binding to ERGO-dependent small RNAs and an increase binding to CSR-class small RNAs (*Figure 6F and G*, *Figure 6—figure supplement 2A*). These data indicate that in the absence of ERGO-dependent small RNAs, some NRDE-3 protein remains associated with CSR-class small RNAs into late embryogenesis. In the *simr-1* and *enri-2* mutants, although we did not observe a significant reduction of NRDE-3-bound ERGO-dependent small RNAs, we saw a significant increase of CSR-class 22G-RNA binding in the *simr-1* mutant and a more modest but still significant increase of CSR-class 22G-RNA binding in the *enri-2* mutant (*Figure 6F and G*, *Figure 6—figure supplement 2B*). These results indicate that SIMR-1 and ENRI-2 are not required for the production of the ERGO-dependent small RNAs during embryogenesis, but may be required for an efficient switch from CSR-class to ERGO-dependent 22G-RNAs.

## NRDE-3 binds to CSR-class 22G-RNAs but does not silence CSR targets in early embryos

We discovered that NRDE-3 unexpectedly binds to CSR-class 22G-RNAs in early embryos, suggesting a potential new role for NRDE-3 that has not been previously reported. We hypothesized that NRDE-3 may function with CSR-1, perhaps to transcriptionally repress germline-expressed genes in early embryos while CSR-1 utilizes its catalytic activity to clear the same maternally-deposited transcripts (*Quarato et al., 2021*). To begin to address this hypothesis, we first sought to assess the degree to which NRDE-3-bound 22G-RNAs are similar to CSR-1-bound 22G-RNAs in early embryos. First, we examined the overlap of NRDE-3-targeted genes in early embryos with CSR-1-targeted genes in embryos or young adult animals. We found that the genes targeted by NRDE-3 substantially overlap with CSR-target genes at both stages (*Figure 7A*; *Quarato et al., 2021*; *Nguyen and Phillips, 2021*). Furthermore, the CSR-target genes yielding the highest abundance of CSR-1-bound small RNAs in embryos also have the highest abundance of NRDE-3-bound small RNAs (*Figure 7B*). These CSR-target genes with highly abundant CSR-bound small RNAs are highly enriched by NRDE-3 only in embryos and not in young adults (*Figure 7—figure supplement 1A*). Next, CSR-class 22G-RNAs tend to be enriched at the 3' ends of mRNAs while WAGO-class 22G-RNAs are more evenly distributed across the gene bodies in adult animals (*Ishidate et al., 2018*; *Singh et al., 2021*). Comparing NRDE-3-bound small RNAs from early embryos to a published dataset of CSR-1-bound small RNA from mixed-stage embryos, we found that both NRDE-3 and CSR-1 are heavily enriched for small RNAs derived from the 3' ends of CSR-target genes in embryos (*Figure 7C*). Interestingly, in adult animals, CSR-1-bound 22G-RNAs are still enriched for small RNA derived from the 3' end of CSR-target genes, however there is additionally a much higher enrichment of small RNAs derived from the gene bodies compared to in embryos (*Figure 7C*). It has previously been proposed that two types of CSR-class 22G-RNAs exist, those that depend on CSR-1 catalytic activity for their production and are derived primarily from target gene bodies, and those that are produced independently of CSR-1 catalytic activity and are derived primarily from target 3' UTRs (*Singh et al., 2021*). Our data points to both NRDE-3 and CSR-1 binding only the latter, CSR-1 catalytic activity-independent, type of CSR-class 22G-RNA in early embryos. In contrast, NRDE-3 does not show enrichment for small RNAs derived from the 3' ends of ERGO target genes in embryos, and rather the small RNAs are distributed more evenly across the gene bodies (*Figure 7—figure supplement 1B*). CSR-1 utilizes its catalytic activity to slice and clear maternally inherited mRNAs from early embryos, preferentially binding to transcripts degraded early in embryogenesis (*Quarato et al., 2021*). We further demonstrate that NRDE-3 similarly binds preferentially to early-degraded transcripts (*Figure 7—figure supplement 1C*). Lastly, the expression of mRNAs targeted by CSR-1 decreases across embryonic development as CSR-1 actively slices and clears these maternal transcripts (*Quarato et al., 2021*). We similarly find that the mRNAs targeted by NRDE-3 in young embryos, which correspond primarily to CSR-target mRNAs, decrease in expression across development, while its targets in young adults, corresponding primarily to ERGO-target mRNAs, increase in expression across development (*Figure 7—figure supplement*

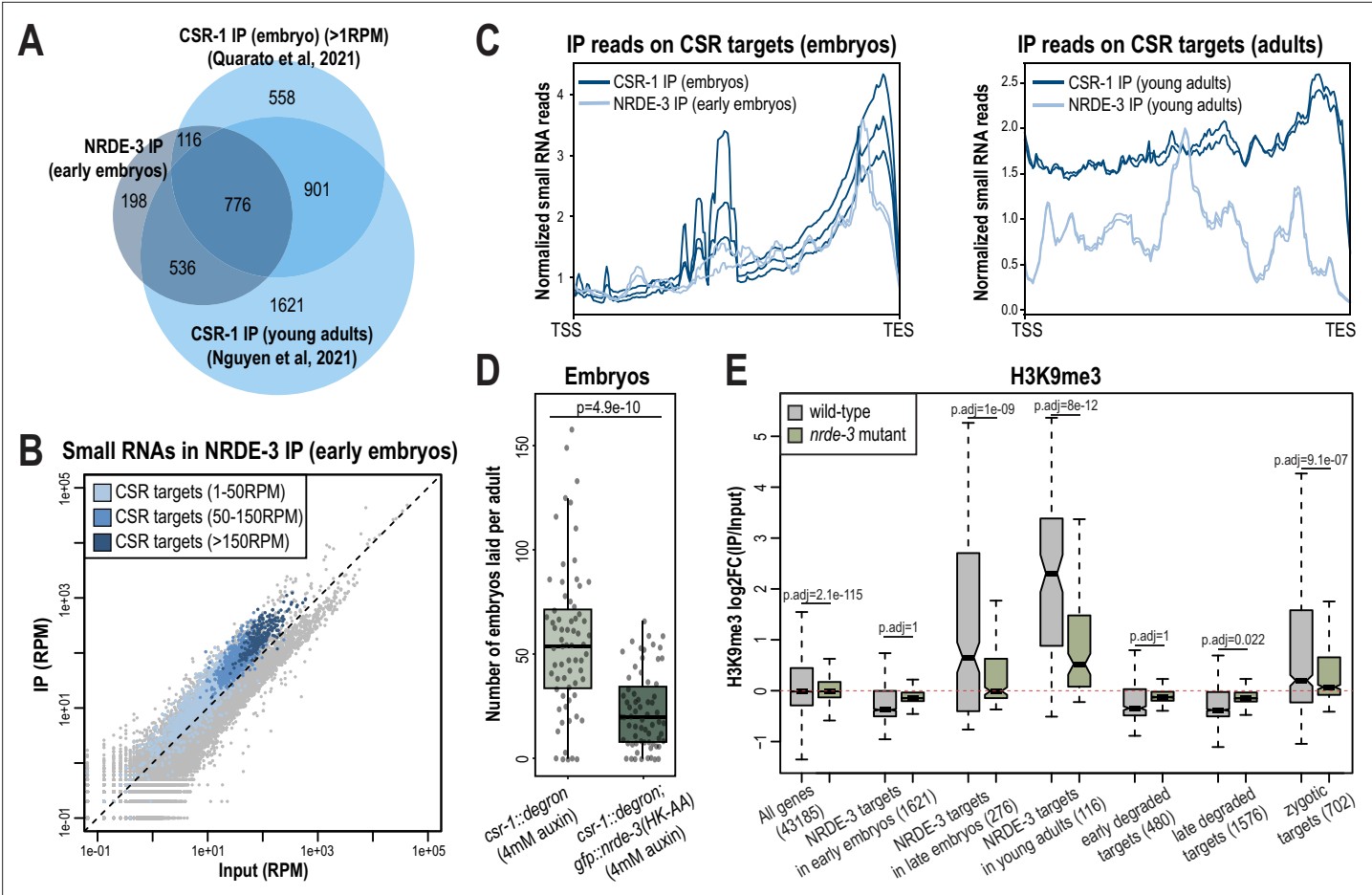

**Figure 7.** NRDE-3 associates with CSR-class 22G-RNAs in early embryos. (**A**) Venn diagram indicates overlap of NRDE-3 IP enriched targets in early embryos (this work), CSR-1 IP enriched targets in young adults (*Nguyen and Phillips, 2021*), and CSR-1 IP enriched targets in embryos (*Quarato et al., 2021*). (**B**) Normalized NRDE-3 IP compared to input small RNA reads in early embryos. CSR-target genes with 1–50 RPM, with 50–100 RPM, and with more than 150 RPM are indicated in light blue, medium blue, and dark blue, respectively. One representative replicate is shown. (**C**) Density plots of small RNA enrichment on CSR targets in CSR-1 IP (dark blue), NRDE-3 IP (light blue) in embryos (left) and adults (right). Transcription start site (TSS) to transcription end site (TES) were plotted using normalized small RNA reads. All replicates are shown as individual lines. (**D**) Box plot quantifying the number of embryos laid per adult *csr-1::degron* or *csr::degron, gfp::nrde-3(HK-AA)* animal on 4 mM auxin plate. At least 65 individuals from each strain were scored. Each dot represents an individual animal, and all data points are shown. (**E**) Box plot depicting log$_2$(fold change of H3K9me3 level in IP vs input) in wild-type (grey) and *nrde-3* mutant (green) mixed-stage embryos, indicating that the H3K9me3 level of NRDE-3 targets in early embryos are not affected in *nrde-3* mutant. Anti-H3K9me3 ChIP-seq data was obtained from *Padeken et al., 2021*. For box plots in D-E, bolded midline indicates median value, box indicates the first and third quartiles, and whiskers represent the most extreme data points within 1.5 times the interquartile range, excluding outliers. Two-tailed t-tests were performed to determine statistical significance and p-values were adjusted for multiple comparisons.

The online version of this article includes the following source data and figure supplement(s) for figure 7:

**Source data 1.** This file contains the raw data used to generate the graph of number of embryos laid shown in *Figure 7D*.

**Figure supplement 1.** NRDE-3 associates different classes of small RNAs during development.

---

1D). Together, these data reveal that NRDE-3 binds to the same group of small RNAs as CSR-1 in early embryos.

To further investigate whether NRDE-3 and CSR-1 function synergistically, we examined the fertility of the *csr-1::degron* strain and the *csr-1::degron; gfp::nrde-3(HK-AA)* strain upon auxin treatment to deplete CSR-1. As expected, both strains had 100% viable progeny with ethanol control treatment (*Figure 7—figure supplement 1E*). When growing on 4 mM auxin plates, the number of embryos laid by the *csr-1::degron; gfp::nrde-3(HK-AA)* double mutant was significantly lower compared to the *csr-1::degron* single mutant and more of the double mutant produced no embryos (11.9%) compared to the *csr-1::degron* single mutant strain (7.5%), indicating a more severe

sterility defect in the *csr-1::degron; nrde-3(HK-AA)* double mutant compared to the *csr-1::degron* alone (**Figure 7D**, **Figure 7—figure supplement 1E**). Additionally, 5.9% of the auxin-treated *csr-1::degron* animals produced some F1 progeny that hatched, compared to no F1 hatching for any of the auxin-treated *csr-1::degron; gfp::nrde-3(HK-AA)* double mutant animals (**Figure 7—figure supplement 1E**). All together, these results indicate that loss of NRDE-3 enhances the fertility defects of CSR-1.

NRDE-3 is a nuclear Argonaute protein that recruits histone methyltransferases to target genes to deposit histone modifications such as H3K9me3 and H3K27me3 at these loci (**Guang et al., 2008**; **Burton et al., 2011**; **Mao et al., 2015**). To examine whether NRDE-3 promotes deposition of H3K9me3 at CSR-target genes during embryogenesis, we analyzed the published anti-H3K9me3 ChIP-seq data of wild-type and *nrde-3* mutant mixed-staged embryos (**Padeken et al., 2021**). In wild-type embryos, the targets of NRDE-3 in young adults, which correspond to ERGO-target genes, have high H3K9me3 levels, and are significantly decreased in the *nrde-3* mutant (**Figure 7E**). These data are consistent with previous research demonstrating that NRDE-3 deposits H3K9me3 at ERGO target genes (**Burton et al., 2011**). However, NRDE-3 targets in early embryos do not show H3K9me3 enrichment in wild-type and do not have a significant change in the *nrde-3* mutant (**Figure 7E**). The same trend is also observed in the early degraded and late degraded targets (**Figure 7E**). These results indicate that the CSR targets are not H3K9 trimethylated in the early embryos. However, we cannot rule out the possibility that NRDE-3 may function to deposit other histone modification targets such as H3K27me3 and H3K23me3 or inhibit RNA Polymerase II (Pol II) on CSR targets to transcriptionally silence these genes in early embryos.

## NRDE-3 associates with CSR-class 22G-RNAs in oocytes

In addition to being expressed in early embryos, NRDE-3 is also expressed in germ cells, starting in late pachytene through oogenesis (**Seroussi et al., 2023**; **Chen and Phillips, 2024b**). Because we were unable to detect a change in H3K9me3 in early embryos in the *nrde-3* mutant, we next sought to determine whether NRDE-3 binds to CSR-class 22G-RNAs in germ cells. Curiously, in late embryos from *eri-1* mutant and *rde-3* mutant, we have observed nuclear localization of NRDE-3 only in the primordial germ cells (**Figure 2A**), raising an intriguing hypothesis that NRDE-3 might bind to CSR-class 22G-RNAs in germ cells throughout development and inherit NRDE-3-bound CSR-class 22G-RNAs to early embryos.

To determine whether NRDE-3 also binds to CSR-class 22G-RNAs in oocytes, we first asked whether the nuclear localization of NRDE-3 in the adult germline depends on ERI-1 and RDE-3, key components of the ERI and *Mutator* complexes, respectively (**Figure 1—figure supplement 1A–B**). Loss of NRDE-3 nuclear localization in these mutants would indicate that NRDE-3 binds to ERGO-dependent 22G-RNAs in oocytes. Instead, we found that NRDE-3 localizes to the nuclei of oocytes in *eri-1* and *rde-3* mutants, similar to wild-type, but is restricted to cytoplasm in the *nrde-3(HK-AA)* small RNA binding mutant (**Figure 8A**). These data demonstrate that NRDE-3 nuclear localization remains dependent on small RNA binding, but ERGO-dependent 22G-RNAs are not required, consistent with NRDE-3 localization in early embryos. Next, to more directly address whether NRDE-3 binds to CSR-class 22G-RNAs in oocytes, we utilized the auxin-inducible degron (AID) system to deplete the RdRP EGO-1 by growing the worms on 4 mM auxin plates starting at the L1 stage (**Zhang et al., 2015**). Surprisingly, NRDE-3 still localizes to nuclei in both germ cells and early embryos upon EGO-1 depletion (**Figure 8B**), indicating that NRDE-3 either does not exclusively bind CSR-class 22G-RNAs in the germline, or NRDE-3 has the capacity to bind other small RNAs when the CSR-class 22G-RNAs are absent. We did observe some cytoplasmic NRDE-3 granules in a subset of 8 cell stage embryos following EGO-1 depletion (**Figure 8B**), suggesting that a proportion of NRDE-3 might be unloaded. To further probe which small RNAs NRDE-3 binds to in the germline, we introduced a *rde-3* mutation into the GFP::NRDE-3; degron::EGO-1 strain to deplete all WAGO-class 22G-RNAs, which include the ERGO-dependent 22G-RNAs, along with the CSR-class 22G-RNAs. We observed that NRDE-3 no longer localizes to the nucleus in both germline and early embryos in the absence of both WAGO-class and CSR-class 22G-RNAs (**Figure 8B**). These results, in combination with our sequencing data, indicate that NRDE-3 likely binds CSR-class 22G-RNAs in the germline and early embryos but has the capacity to additionally bind WAGO-class 22G-RNAs when CSR-class 22G-RNAs are depleted. Furthermore, because somatic transcription is not initiated in early embryos until the 4 cell stage

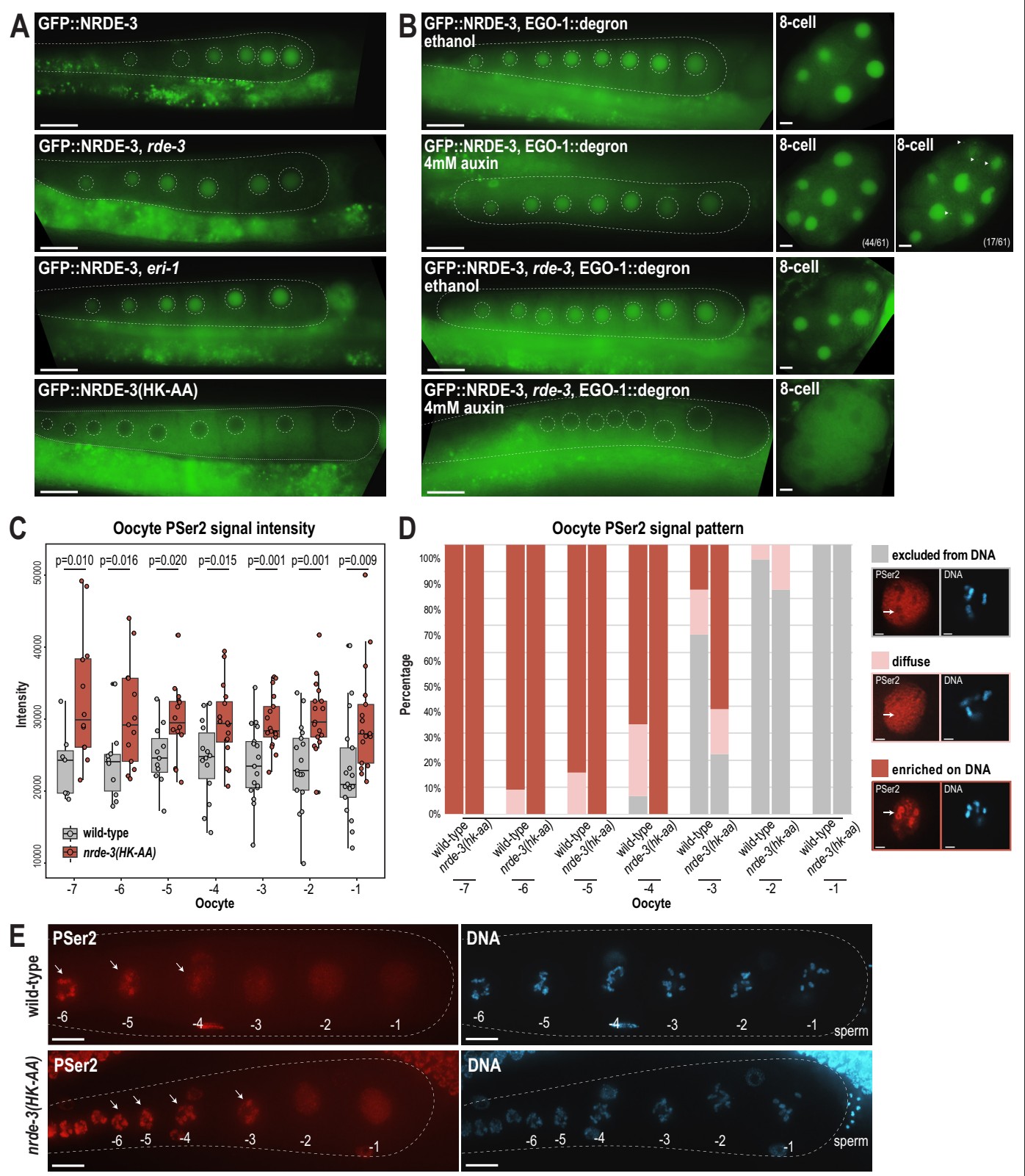

**Figure 8.** NRDE-3 associates with CSR-class 22G-RNAs to represses RNA Polymerase II in oocytes. (**A**) Live imaging of GFP::3xFLAG::NRDE-3 in 1-day-adult germlines for wild-type, *eri-1*, *rde-3*, and *nrde-3(HK-AA)* mutants, showing that NRDE-3 localizes to the nuclei of oocytes in wild-type, *eri-1* mutant, and *rde-3* mutants, and to the cytoplasm in the *nrde-3(HK-AA)* mutant. At least five individual gonads were imaged for each genotype. Dotted white line traces the proximal portion of the *C. elegans* gonad and outlines the individual oocytes. Scale bars, 25 μm. (**B**) Live imaging of 1-day-adult germlines

*Figure 8 continued on next page*

*Figure 8 continued*

and 8 cell embryos for EGO-1::degron; GFP::3xFLAG::NRDE-3 (top) and EGO-1::degron; GFP::3xFLAG::NRDE-3 in a *rde-3* mutant (bottom) with ethanol (control) and 4 mM auxin treatment, showing that loss of both WAGO-class and CSR-class 22G-RNAs (*rde-3* mutant and degron-mediated EGO-1 depletion) is necessary to result in cytoplasmic localization of NRDE-3 in both oocytes and early embryos. At least five individual gonads and embryos were imaged for each treatment condition. Dotted white line traces the proximal portion of the *C. elegans* gonad and outlines the individual oocytes. Arrowheads indicate granule localized NRDE-3 in 8 cell stage embryos. Scale bars, 25 μm in adults and 5 μm in embryos. (**C**) Box plot quantifying the RNA Pol II PSer2 signal intensity in oocytes of wild-type (GFP::NRDE-3 strain), and *nrde-3(HK-AA)* mutant (GFP::NRDE-3(HK-AA) strain), showing that PSer2 signal is significantly increased in all oocytes in the *nrde-3(HK-AA)* mutant. Each dot represents an individual oocyte, and all data points are shown. Bolded midline indicates median value, box indicates the first and third quartiles, and whiskers represent the most extreme data points within 1.5 times the interquartile range. Two-tailed t-tests were performed to determine statistical significance. See Materials and methods for a detailed description of quantification methods. (**D**) Bar plot quantifying the RNA Pol II PSer2 expression pattern in wild-type (GFP::NRDE-3 strain), and *nrde-3(HK-AA)* mutant (GFP::NRDE-3(HK-AA) strain) oocytes, showing that the PSer2 signal is retained on DNA longer in the *nrde-3(HK-AA)* mutant. At least 10 oocytes were used for quantification for each strain. Examples of three patterns of PSer2 signal are shown on right. Arrows point to a region of DNA to highlight PSer2 enrichment or exclusion. Scale bars, 2 μm. (**E**) Immunofluorescence imaging of PSer2 signal and DAPI stained DNA in oocytes of wild-type (GFP::NRDE-3), and *nrde-3(HK-AA)* mutant (GFP::NRDE-3(HK-AA)), showing that the PSer2 signal appears earlier on DNA in the *nrde-3(HK-AA)* mutant. Images are maximum intensity projections of 12.5 μm z-stack, which allows for optimal visualization of the DNA-associated PSer2 signal, but obscures the 'excluded from DNA' pattern. At least five individual animals for each genotype. Arrows indicate the PSer2 signal on DNA. Scale bars, 25 μm.

The online version of this article includes the following source data for figure 8:

**Source data 1.** This file contains the raw data used to generate the graphs of oocyte PSer2 signal intensity and pattern number shown in ***Figure 8C–D***.

(***Seydoux and Fire, 1994***), we conclude that the NRDE-3 is loaded with CSR-class 22G-RNAs in the parental germline and then transmitted to early embryos.

## NRDE-3 represses RNA Pol II and promotes global transcriptional repression in oocytes

Oocytes undergo global transcriptional repression across diverse species (***Woodland, 1987***). In *C. elegans*, transcriptional shutdown begins during the diakinesis stage of late oocytes to prepare for fertilization, and reactivates at the 4 cell stage embryos to initiate zygotic transcription (***Walker et al., 2007***). Previous work has shown that topoisomerase II acting with the condensin II complex, H3K9me3, and the zinc-finger containing protein PIE-1 are critical for global transcriptional repression (***Belew et al., 2023***), but the complete molecular mechanisms underlying the oocyte-to-embryo transition remain not fully understood. We therefore aimed to address whether germline-expressed NRDE-3 could play a role in these pathways.

Our hypothesis that NRDE-3 may mediate transcriptional repression in oocytes is supported by several lines of evidence. First, NRDE-3 is expressed at the correct place and time—in the germline, from late pachytene to diakinetic oocytes (***Seroussi et al., 2023***)—and is known to co-transcriptionally silencing genes by depositing H3K9me3 and inhibiting Pol II transcription (***Guang et al., 2008***). Second, the slicing activity of CSR-1 is required to inhibit Pol II-dependent transcription in maturing oocytes (***Fassnacht et al., 2018***). CSR-1 is not known to directly inhibit transcription, but it is required for the production of at least a subset of the CSR-class 22G-RNAs (***Singh et al., 2021***); therefore, the CSR-1-dependent Pol II inhibition could be mediated by NRDE-3-bound CSR-class 22G-RNAs. Third, H3K9me3 and the H3K9me3 methyltransferase SET-25 are required for transcription repression in oocytes (***Belew et al., 2023***). NRDE-3 can act to recruit SET-25 and establish H3K9me3-containing heterochromatin (***Padeken et al., 2021***), making it a strong candidate for initiating H3K9me3-dependent transcriptional silencing in oocytes.

The above evidence connects Pol II repression in oocytes to the co-transcriptional silencing function of NRDE-3 and the requirement for CSR-class 22G-RNAs in oocytes. Phosphorylation of RNA Polymerase II (Pol II) on its large subunit C-terminal domain (CTD) serves as a well-studied marker of transcription activity, with Ser5 phosphorylation indicating transcription initiation and Ser2 phosphorylation marks elongation. Therefore, to monitor Pol II activity in oocytes, we stained wild-type and *nrde-3(HK-AA)* mutant gonads using an antibody specific for phosphorylated serine 2 (PSer2) on the Pol II C-terminal domain (CTD), a marker for transcription elongation (***Belew et al., 2023***; ***Seydoux and Dunn, 1997***). Consistent with previously reports, PSer2 staining in the most proximal oocytes exhibited distinct patterns of localization: (1) PSer2 excluded from DNA ('excluded from DNA'), (2) PSer2 present both on and off DNA ('diffuse'), and (3) PSer2 enriched on DNA ('enriched on DNA').

The PSer2 signal intensity and the localization pattern serve as indicators of active transcription levels (*Belew et al., 2023*). Quantification the PSer2 signal intensity shows a significant increase in PSer2 in the *nrde-3(HK-AA)* mutant across all oocytes (from –1 to –7), indicating a global increase in transcription elongation in the *nrde-3(HK-AA)* mutant (*Figure 8C*). Furthermore, by quantifying the PSer2 localization patterns, we found that elongating RNA Pol II stays associated with DNA longer in the *nrde-3(HK-AA)* mutant, compared to wild-type (*Figure 8D–E*). This data is consistent with what has been observed previously in the *set-25* mutant (*Belew et al., 2023*). Therefore, we conclude that NRDE-3 contributes to global transcriptional repression in oocytes by repressing RNA Pol II.

## Discussion

Germ granules are phase-separated condensates that localize to the perinuclear region of germ cells. In *C. elegans*, the known constituents of the germ granule have expanded over the last decades, such that germ granules now comprise multiple domains including P granules, *Mutator* foci, Z granules, SIMR foci, E granules, and D granules. Here, we discovered that several components of SIMR foci and *Mutator* foci also localize to cytoplasmic granules during specific stages of embryogenesis, where their temporal expression is regulated by autophagic degradation. We propose that these granules may serve as sites for the synthesis and loading of 22G-RNAs into the nuclear Argonaute NRDE-3. Furthermore, we showed that NRDE-3 switches its small RNA targets during embryogenesis, coincident with the formation of SIMR granules; during oogenesis, NRDE-3 binds to CSR-class 22G-RNAs to promote global transcriptional repression prior to the oocyte-to-embryo transition, and then in the developing soma, NRDE-3 binds to ERGO-dependent 22G-RNAs to silence retrotransposons and recently duplicated genes. Together, our study reveals a new world of embryonic RNAi factor condensates and uncovers two temporally distinct roles for NRDE-3, underscoring the need for careful examination of localization and targets of RNAi pathways across development (*Figure 9*).

### A role for SIMR-1 as a platform for nuclear Argonaute protein loading

Previously, we demonstrated that SIMR-1 and HRDE-2 are required to recruit unloaded HRDE-1, the germline nuclear Argonaute protein, to germ granules and to ensure correct 22G-RNA loading (*Chen and Phillips, 2024a*). Here, we reveal that SIMR-1 and another HRDE-2 paralog, ENRI-2, are similarly essential to recruit unloaded NRDE-3, the somatic nuclear Argonaute protein, to embryonic SIMR granules. We speculate that SIMR-1 and ENRI-2 are similarly important for NRDE-3 22G-RNA loading; however, we did not observe a significant change in the ERGO-dependent 22G-RNAs loaded by NRDE-3 in *simr-1* or *enri-2* mutant embryos compared to wild-type. While initially surprising based on the results of similar experiments with HRDE-1 in the germline, we envision several possible explanations. First, it is possible that SIMR-1 and ENRI-2 act to bring unloaded NRDE-3 in close proximity to the ERGO-dependent 22G-RNA biogenesis machinery, but that NRDE-3 loading can still occur diffusely in the cytoplasm, albeit with lower efficiency. Both the RdRP RRF-1 and unloaded NRDE-3 diffusely localize to cytoplasm in the *simr-1* mutant (*Figures 3A and 4D*), suggesting that NRDE-3 may load the ERGO-class small RNAs synthesized in the cytoplasm in the absence of SIMR-1. Differences in NRDE-3 loading efficiency would likely not be detected by our NRDE-3 IP-small RNA sequencing experiment. Second, SIMR-1 and ENRI-2 could act to sequester unloaded NRDE-3 away from other small RNAs (i.e. CSR-class 22G-RNAs) to prevent misloading. Misloading should be detectable in our NRDE-3 IP-small RNA sequencing experiment and we do see an increase in CSR-class 22G-RNAs loaded by NRDE-3 in late embryos; however, it is unclear the extent to which newly synthesized, and unloaded CSR-class 22G-RNAs are even present in the cytoplasm, as the primary source for CSR-class 22G-RNAs may be the maternal germline. Thus, unlike in the adult germline where HRDE-1 incorrectly loads CSR-class 22G-RNAs in the absence of HRDE-2, there may not be an equivalent source of incorrect small RNAs that NRDE-3 can bind to (i.e. correct length, 5' nucleotide and modifications) in the embryo. To further probe these possibilities, we need to more carefully assess the dynamics of NRDE-3 loading across embryonic development and possibly disrupt the formation of embryonic CSR granules to determine whether compartmentalization of the CSR-class 22G-RNA pathway is also contributing to correct loading of NRDE-3 in the absence of SIMR-1 and ENRI-2.

We do not know the precise functions of SIMR-1 and ENRI-2; however, we have previously proposed that SIMR-1 mediates protein-protein interactions through its extended Tudor domain

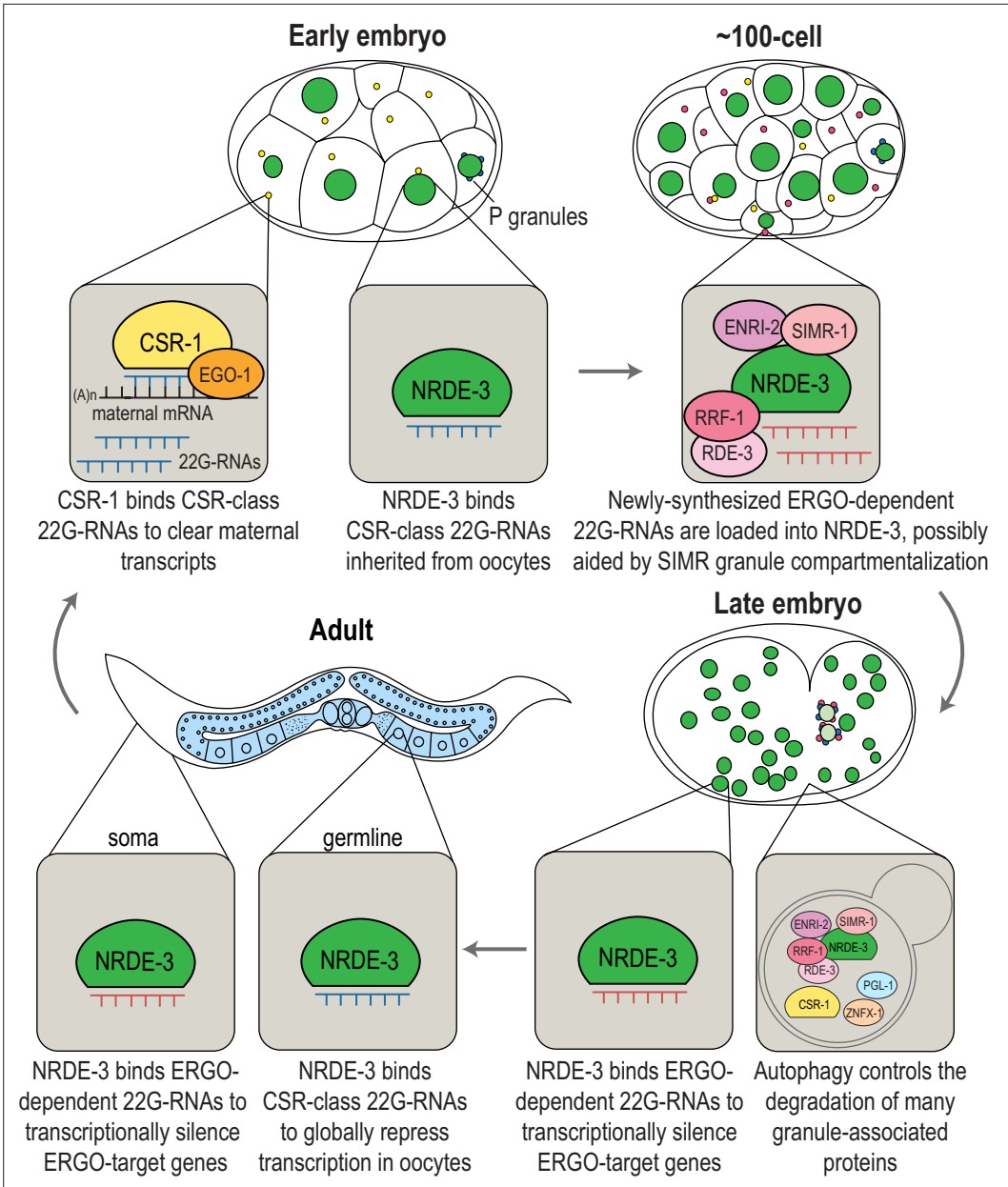

**Figure 9.** Model for temporally- and developmentally-regulated NRDE-3 function. Model of NRDE-3, SIMR-1, and CSR-1 function during *C. elegans* development. In early embryos, CSR-1 and EGO-1 localize to CSR granules and synthesize CSR 22G-RNAs to slice and clear maternal mRNAs. NRDE-3 binds CSR 22G-RNA in the nucleus, which are inherited from the oocytes. During mid-embryogenesis (e.g. around the 100 cell stage), unloaded NRDE-3, ENRI-2, RRF-1, and RDE-3 localize to SIMR granules in somatic cells dependent on SIMR-1, where we propose that ERGO-dependent 22G-RNAs are produced and loaded into NRDE-3. In late embryos, NRDE-3 binds ERGO-dependent 22G-RNAs and silences ERGO-target genes in the nucleus, while autophagy controls selective degradation of SIMR and other embryonic granules. In adult *C. elegans*, somatic localized NRDE-3 associates with ERGO-dependent 22G-RNAs to transcriptionally silence ERGO-target genes, while germline localized NRDE-3 associates with CSR-class 22G-RNAs to globally repress transcription and promote chromatin compaction in oocytes, ultimately being deposited into early embryos.

The online version of this article includes the following figure supplement(s) for figure 9:

**Figure supplement 1.** Predicted structures of SIMR-1, nuclear Argonaute proteins, and interactors.

(*Manage et al., 2020*). ENRI-2 and its paralog HRDE-2 have structural similarities to a HELICc domain, and SIMR-1, ENRI-2, and HRDE-2 have large unstructured domains (*Figure 9—figure supplement 1A*; *Lewis et al., 2020*; *Chen and Phillips, 2024a*). With the advent of protein complex prediction algorithms (*Abramson et al., 2024*), we sought to examine the potential physical interactions between ENRI-2 and NRDE-3, as well as their paralogs HRDE-2 and HRDE-1. In both models, the structured HELICc domains of HRDE-2 and ENRI-2 dock on the Mid domains of their respective Argonaute partners. Interestingly, the unstructured C-terminal domains of ENRI-2 and HRDE-2 extend into the small RNA binding pocket of their respective nuclear Argonaute binding partners (*Figure 9—figure supplement 1B*). At this point, we do not know whether these structures are reflective of the actual geometry of the proteins in vivo, but it is tempting to speculate that the C-terminal disordered regions of the ENRI-2/HRDE-2 proteins could regulate 22G-RNA loading through interaction with the small RNA binding pockets of NRDE-3 and HRDE-1. Further study will be necessary to determine whether these interactions between disordered region and the small RNA binding pocket are necessary for correct small RNA loading and whether that mechanism extends to other WAGO proteins.

## Compartmentalization of RNAi pathways

Most of the studies on the organization of *C. elegans* RNAi factors in granules focus on the germline. Here, we find that multiple proteins associated with 22G-RNA biogenesis and function, including SIMR-1, RDE-3, RRF-1, ENRI-2, and unloaded NRDE-3, are localized to distinct condensates in *C. elegans* embryos. We speculate that these SIMR granules, which appear and then disappear during the course of embryonic development, play a functional role in the NRDE-3 nuclear RNAi pathway. This idea leads to an intriguing question: what role does organization of the RNAi pathways into condensates play in the soma vs. in the germline?

In germ cells, RNAi factors are visibly segregated into distinct compartments within the germ granule which assemble hierarchically (*Uebel et al., 2023*). Germ granules are also intimately linked to nuclear pores, leading to a model where highly concentrated mRNAs, newly exported from and adjacent to the nuclear pore, nucleate assembly of regulatory factors into visible granules. In fact, nuclear pores are clustered beneath germ granules in germ cells, and evidence suggests that most, if not all, nascent mRNAs are exported through pores associated with germ granules (*Pitt et al., 2000*; *Sheth et al., 2010*). In contrast, nuclear pores are distributed more evenly across the nuclear periphery in embryos and, in this work, we find that while some embryonic SIMR granules appear adjacent to the nuclear periphery, many are distributed in the cytoplasm (see *Figure 2A*, for example). Thus, unlike in germ granules, there is no obvious trajectory from the nucleus that RNAs would follow to end up in embryonic SIMR granules. Further, in the germline, we have speculated that the adjacent and hierarchical assembly of germ granule compartments could be determined by the order of molecular events required for RNA silencing (*Uebel et al., 2023*). While we on occasion see docking between embryonic SIMR granules and CSR granules (*Figure 4I*), we do not see any more complex arrangement of granule compartments in embryos similar to what we have observed in the germline. What that means regarding the functionality of embryonic SIMR granules is unclear. Another possibility worth considering is that SIMR granules are not actually required for ERGO-dependent 22G-RNA biogenesis and NRDE-3 loading in embryos but rather that they reflect a concentration of the small RNA biogenesis machinery beyond the solubility limit of the cytoplasm, resulting in the demixing of some RNP complexes into visible SIMR granules (*Putnam et al., 2023*). By this 'incidental condensate' model, ERGO-dependent 22G-RNA biogenesis and NRDE-3 loading occur just as efficiently, or perhaps more so, diffusely in the cytoplasm.

Both embryonic and germ granules exhibit dynamic expression patterns, suggesting that expression and function of small RNA factors are critical at discrete developmental time points. In the germline, multiple Argonaute proteins are expressed exclusively during oogenesis (ERGO-1) or spermatogenesis (ALG-3, ALG-4, CSR-1b, WAGO-10) (*Billi et al., 2012*; *Han et al., 2009*; *Conine et al., 2010*; *Reinke et al., 2004*; *Nguyen and Phillips, 2021*; *Charlesworth et al., 2021*); and MUT-16 expression fluctuates across germ cell development, peaking in the mitotic region (*Uebel et al., 2020*). Similarly, embryonic SIMR granules appear in early embryos and disappear by late embryogenesis. Regardless as to whether SIMR granules are incidental condensates or functional sites for NRDE-3 loading, these data indicate that the levels or activities of these proteins are developmentally regulated.

It is additionally curious that embryonic and germ granules share many protein components yet possess distinct differences in content and assembly requirements. For instance, several RNAi proteins, such as RRF-1 and RDE-3, are shared between *Mutator* foci and embryonic SIMR granules, while the paralogous ENRI-2/NRDE-3 and HRDE-2/HRDE-1 pairs are found in embryonic SIMR granules and germline SIMR foci, respectively. It is unclear why the *Mutator* and SIMR components are visible as separate compartments in germ granules but are together in embryonic SIMR foci. This difference is highlighted by the requirement for MUT-16 in the assembly of embryonic SIMR granules but not germline SIMR foci (*Figure 4E*; *Manage et al., 2020*). Further investigation into the assembly and protein components of embryonic and germ granules will be crucial for elucidating the functional differences between embryonic and germ granules and dissecting the mechanisms of 22G-RNA loading into NRDE-3.

## Spatial-temporal regulation of SIMR and other embryonic granules by autophagy

Autophagy is a conserved eukaryotic protein degradation pathway involving the formation of double-membrane autophagosomes, which fuse with lysosomes for degradation (*Nakatogawa et al., 2009*). Autophagy has been linked to small RNA pathways and germ granules in *C. elegans* and other organisms. In *Arabidopsis*, the turnover of Argonaute protein AGO1 is controlled by autophagy in both antiviral and nonviral contexts (*Derrien et al., 2012*). In mammalian cells, AGO2 and Dicer are targeted for degradation via autophagy (*Gibbings et al., 2012*). In *C. elegans*, P granule components PGL-1 and PGL-3 are recruited to somatic aggregates during embryogenesis and degraded by autophagy (*Zhang et al., 2009*). Interestingly, miRNA pathway components ALG-1, ALG-2 and their interactor AIN-1/GW182 are also degraded by autophagy, while in aggregates distinct from PGL somatic aggregates (*Zhang and Zhang, 2013*).

Here, we discovered that, similar to P granule components, the expression of many other granule-associated proteins in *C. elegans*, including the core Z granule protein ZNFX-1, the D granule protein CSR-1, and the SIMR granule protein SIMR-1, are also controlled by autophagy during embryogenesis. It is worth noting that the timing of degradation differs amongst the different granules, with P granules and Z granules being cleared from somatic cells within the first few cell divisions of the developing embryo (*Strome and Wood, 1983*; *Wan et al., 2018*), while CSR granules are removed between 28- and 100 cell stage (*Figure 4G*), and SIMR granules disappear between 100- and 200 cell stage of embryonic development (*Figures 1C and 2C*). These results suggest an intriguing hypothesis that different granules might be produced independently at designated developmental stages and cytosolic locations to perform specific functions, and later actively degraded by autophagy when not needed. Future studies will be needed to carefully examine the role and fate of each type of granule during embryogenesis to test this hypothesis.

Lastly, while autophagy pathway plays an important role in regulating expression of many granule components, we cannot role the possibility that the ubiquitin-proteasome system may also contribute to protein turnover in the small RNA pathway. In fact, *Zhang et al., 2009* demonstrated that, unlike PGL-1 and PGL-3, the other P granule components GLH-1 and GLH-4 are not degraded via autophagy. Additionally, *DeRenzo et al., 2003* showed that germline-expressed CCCH figure proteins, including PIE-1, POS-1, MEX-1, are selectively targeted for degradation in somatic cells by the E3 ubiquitin ligase subunit-interacting protein ZIF-1 through the ubiquitin-proteasome system. Furthermore, previous research suggest that the proteosome and autophagy pathways can act as compensatory mechanisms under certain contexts (*Ji and Kwon, 2017*). Future studies will be needed to interrogate the roles of these two protein degradation pathways in regulating protein turnover in the small RNA pathway.

## The small RNA plasticity of NRDE-3

Argonautes are conventionally known to bind small RNAs with high specificity. In this study, we unveil the remarkable versatility of the nuclear Argonaute NRDE-3, demonstrating its ability to bind multiple classes of small RNAs and exhibit distinct functions throughout development. Argonaute proteins with the capacity to bind multiple types or classes of small RNAs have been observed in other organisms. For example, both siRNAs and miRNAs can be loaded into the four human Argonautes (Ago1-4) and both siRNAs and miRNAs can guide Ago2-dependent target cleavage (*Meister et al., 2004*; *Liu*

*et al., 2004*). Our discovery is somewhat different, however, in that NRDE-3 binds its two preferred classes of small RNAs, CSR-class 22G-RNAs and ERGO-dependent 22G-RNAs, at distinct developmental stages, indicating that there must be a switch from one class of small RNA to the other during embryogenesis. Interestingly, a more recent study in the parasitic nematode *Ascaris* revealed that the *Ascaris* paralog of NRDE-3, AsNRDE-3, exhibits a dramatic change in associated small RNAs during spermatogenesis, targeting repetitive sequences and transposons in early stages of spermatogenesis and mRNAs in late meiosis (*Zagoskin et al., 2022*). Curiously, the mRNAs targeted by AsNRDE-3 in late meiosis largely overlap with the targets of AsCSR-1, the *Ascaris* paralog of CSR-1, and it is proposed that AsNRDE-3 could act in concert with AsCSR-1 at the late stages of meiosis to clear spermatogenic and meiotic mRNAs from the developing spermatids (*Zagoskin et al., 2022*). These data further suggest that the ability of NRDE-3 to target both repetitive sequences and germline-expressed genes at distinct developmental timepoints may be a conserved feature of this protein. It is currently unknown how this small RNA switching is achieved. It is possible that there is an active mechanism to unload the CSR-class 22G-RNAs and replace them with ERGO-dependent 22G-RNAs, or to degrade NRDE-3 loaded with CSR-class 22G-RNAs. However, we prefer the simpler model where NRDE-3 loaded with CSR-class 22G-RNAs, initially deposited into embryo from the maternal germline, are diluted out as the animal develops. Newly synthesized NRDE-3 in the embryo is loaded with ERGO-dependent 22G-RNAs to execute the small RNA 'switch'. The idea that Argonaute proteins can be utilized at distinct timepoints with different small RNA partners to create multi-functionality is intriguing, especially in the vein of rapidly clearing transcripts from a cell to engineer a new developmental program. Achieving higher resolution small RNA-Argonaute interactions with tissue- and developmental-specific staging will be crucial to fully elucidate the roles of Argonaute proteins during development in *C. elegans* and other organisms.

In summary, this work investigating the role of SIMR granules in embryos, together with our previous study of SIMR foci in the germline (*Chen and Phillips, 2024a*), has identified a new mechanism for regulating nuclear Argonaute protein localization in *C. elegans*. The two paralogous proteins, HRDE-2 and ENRI-2, recruit unloaded nuclear Argonautes HRDE-1 and NRDE-3 to small RNA production centers organized by SIMR-1, where we speculate that loading can occur. These small RNA loading sites are essential in the germline to promote small RNA binding specificity; however, they may also contribute to efficiency and specificity of small RNA loading in embryos. We further discovered an intriguing repository of cytoplasmic granules during embryogenesis that do not exhibit the same organization or hierarchical assembly as germ granules, but share the same autophagy-mediated degradation as one another. These results highlight the importance of further investigation into the relationship between RNA silencing pathways and RNA granules during embryogenesis. Lastly, we observed a striking phenomenon where the NRDE-3 nuclear Argonaute protein possesses the ability to switch small RNA binding partners, altering mRNA targets and function during development. Together, these findings reveal that the precise regulation of small RNA pathway components through diverse mechanisms, such as spatial-temporal separation and hierarchical physical interactions, is crucial for accurate gene regulation and developmental transitions in *C. elegans*.

## Materials and methods

### Key resources table

| Reagent type (species) or resource | Designation | Source or reference | Identifiers | Additional information |
|---|---|---|---|---|
| Gene (*C. elegans*) | *simr-1* | Wormbase | WBGene00015504 | |
| Gene (*C. elegans*) | *nrde-3* | Wormbase | WBGene00019862 | |
| Gene (*C. elegans*) | *enri-1* | Wormbase | WBGene00018386 | |
| Gene (*C. elegans*) | *enri-2* | Wormbase | WBGene00011981 | |
| Gene (*C. elegans*) | *eri-1* | Wormbase | WBGene00001332 | |
| Gene (*C. elegans*) | *ergo-1* | Wormbase | WBGene00019971 | |
| Gene (*C. elegans*) | *rrf-1* | Wormbase | WBGene00004508 | |

*Continued on next page*

*Continued*

| Reagent type (species) or resource | Designation | Source or reference | Identifiers | Additional information |
|---|---|---|---|---|
| Gene (*C. elegans*) | *rde-3/mut-2* | Wormbase | WBGene00003499 | |
| Gene (*C. elegans*) | *ego-1* | Wormbase | WBGene00001214 | |
| Gene (*C. elegans*) | *csr-1* | Wormbase | WBGene00017641 | |
| Gene (*C. elegans*) | *mut-16* | Wormbase | WBGene00003508 | |
| Gene (*C. elegans*) | *cgh-1* | Wormbase | WBGene00000479 | |
| Gene (*C. elegans*) | *rsd-2* | Wormbase | WBGene00004681 | |
| Gene (*C. elegans*) | *hrde-2* | Wormbase | WBGene00011324 | |
| Gene (*C. elegans*) | *rde-12* | Wormbase | WBGene00010280 | |
| Gene (*C. elegans*) | *rsd-6* | Wormbase | WBGene00004684 | |
| Gene (*C. elegans*) | *znfx-1* | Wormbase | WBGene00014208 | |
| Gene (*C. elegans*) | *pgl-1* | Wormbase | WBGene00003992 | |
| Gene (*C. elegans*) | *lgg-1* | Wormbase | WBGene00002980 | |
| Strain, strain background (*C. elegans*; hermaprhodites) | Bristol N2 | Wormbase | RRID:WB-STRAIN:WBStrain00000001 | |
| Strain, strain background (*E. coli*) | OP50 | *Caenorhabditis* Genetics Center | RRID:WB-STRAIN:WBStrain00041971 | |
| Strain, strain background (*E. coli*) | HT115-L4440 | *Caenorhabditis* Genetics Center | RRID:WB-STRAIN:WBStrain00041080; RRID:Addgene_1654 | control RNAi |
| Strain, strain background (*E. coli*) | HT115- lgg-1 (C32D5.9.1) | *Caenorhabditis* Genetics Center | | Ahringer RNAi library |
| Genetic reagents (*C. elegans*) | List of strains | This study, **Supplementary file 1**; *Caenorhabditis* Genetics Center | | |
| Antibody | Anti-HA 3F10 Peroxidase (Rat monoclonal) | Roche | Cat# 12013819001, RRID:AB_390917 | 1:1000 |
| Antibody | Anti-FLAG M2 (Mouse monoclonal) | Sigma Aldrich | Cat# F1804, RRID:AB_262044 | 1:1000 |
| Antibody | Anti-RNAPIIpSer2 (Rabbit Polyclonal) | Abcam | Cat# ab5095, RRID:AB_304749 | 1:500 |
| Antibody | Anti-Rabbit IgG AlexaFluor 555 (Goat Polyclonal) | Thermo Fisher | Cat# A21429, RRID:AB_2535850 | 1:1000 |
| Antibody | Anti-actin IgG (Mouse monoclonal) | Abcam | Cat# ab3280, RRID:AB_303668 | 1:1000 |
| Antibody | Goat anti-mouse IgG Secondary HRP | Thermo Fisher | Cat# A16078, RRID:AB_2534751 | 1:10,000 |
| Sequence-based reagent | List of oligonucleotides | This study, **Supplementary file 2** | | |
| Sequence-based reagent | High Throughput Sequencing Data (this study) | NCBI GEO | GSE273239 | |
| Sequence-based reagent | High Throughput Sequencing Data | NCBI GEO **Quarato et al., 2021** | GSE146057, GSE146062 | |
| Sequence-based reagent | High Throughput Sequencing Data | NCBI GEO **Padeken et al., 2021** | GSE156548 | |

*Continued on next page*

*Continued*

| Reagent type (species) or resource | Designation | Source or reference | Identifiers | Additional information |
|---|---|---|---|---|
| Sequence-based reagent | High Throughput Sequencing Data | NCBI GEO *Seroussi et al., 2023* | GSE208702 | |
| Commercial assay or kit | Qubit 1 x dsDNA HS Assay Kit | Thermo Fisher | Cat# Q33231 | |
| Chemical compound, drug | Anti-FLAG M2 Affinity Matrix | Sigma Aldrich | Cat# A2220, RRID:AB_10063035 | |
| Chemical compound, drug | TrIzol Reagent (phenol, Guanidine isothiocyanate, Ammonium thiocyanate) | Thermo Fisher | Cat #15596018 | |
| Chemical compound, drug | Pierce Protease Inhibitor Tablets, EDTA-free | Thermo Fisher | Cat# A32965 | |
| Chemical compound, drug | Nonidet P 40 substitute | Sigma Aldrich | Cat# 11332473001 | |
| Chemical compound, drug | SureCast Acrylamide Solution (40%) | Fisher | Cat# HC2040 | |
| Chemical compound, drug | Urea >99.5% Molecular Biology Grade Ultrapure | Thermo Fisher | Cat# J75826.A1 | |
| Chemical compound, drug | TEMED | Thermo Fisher | Cat# 17919 | |
| Chemical compound, drug | Ammonium persulfate | Thermo Fisher | Cat# 17874 | |
| Chemical compound, drug | GeneRuler Ultra Low Range DNA | ThermoFisher | Cat# SM1213 | |
| Chemical compound, drug | Ambion RNA loading buffer II | ThermoFisher | Cat# AM8547 | |
| Chemical compound, drug | SYBR Gold Nucleic Acid Gel Stain | Life Technologies | Cat# S11494 | |
| Chemical compound, drug | Bolt 4–12% Bis-Tris Plus Gels | Thermo Fisher | Cat# NW04122BOX | |
| Chemical compound, drug | Nitrocellulose Pre-Cut Blotting Membranes | Thermo Fisher | Cat# LC2001 | |
| Chemical compound, drug | RNA 5' Polyphosphatase | EpiCentre | Cat# RP8092H | |
| Chemical compound, drug | Phenol/Chloroform/Isoamyl Alcohol (pH 4.3) | Fisher | Cat# BP1754I-100 | |
| Chemical compound, drug | T4 RNA Ligase 2, truncated KQ | New England Biolabs | Cat# M0373L | |
| Chemical compound, drug | T4 RNA Ligase 1 | New England Biolabs | Cat# M0204L | |
| Chemical compound, drug | SuperScript III | Thermo Fisher | Cat# 18080–051 | |
| Chemical compound, drug | Q5 High Fidelity DNA Polymerase | New England Biolabs | Cat# M0491L | |
| Software, algorithm | Adobe Photoshop | Adobe | RRID:SCR_014199 | |
| Software, algorithm | Adobe Illustrator | Adobe | RRID:SCR_014199 | |
| Software, algorithm | FIJI/ImageJ2/Fiji is just ImageJ (version 2.9.0) | *Schindelin et al., 2012* | RRIF:SCR_002285 | |
| Software, algorithm | FASTX-Toolkit | Hannon Lab - CSHL | RRID:SCR_005534 | |
| Software, algorithm | Bowtie2 (Version 2.2.2) | *Langmead and Salzberg, 2012* | RRID:SCR_005476 | |
| Software, algorithm | DESeq2 (Version 1.22.2) | *Love et al., 2014* | RRID:SCR_015687 | |
| Software, algorithm | edgeR (Version 3.40.2) | *Robinson et al., 2010* | RRID:SCR_012802 | |
| Software, algorithm | Trimmomatic (Version 0.36) | *Bolger et al., 2014* | RRID:SCR_011848 | |
| Software, algorithm | BioVenn | *Hulsen et al., 2008* | | |
| Software, algorithm | Integrated Genomics Viewer | *Robinson et al., 2010* | RRID:SCR_011793 | |

## *C. elegans* stains

*C. elegans* strains were maintained at 20 °C on NGM plates seeded with OP50 *E. coli* according to standard conditions unless otherwise stated (*Brenner, 1974*). All strains used in this project are listed

in *Supplementary file 1*.

## CRISPR-mediated strain construction

For *nrde-3(cmp324[HK-AA])*, *enri-1(cmp328)*, *enri-2(cmp318)*, and *rde-3/mut-2(cmp337)*, we used an oligo repair template and RNA guide. For *enri-1(cmp320[enri-1::mCherry::2xHA])*, we used an RNA guide and PCR amplified repair template (*Supplementary file 2*). For injections using a single gene-specific crRNA, the injection mix included 0.25 µg/µl Cas9 protein (IDT), 100 ng/µl tracrRNA (IDT), 14 ng/µl dpy-10 crRNA, 42 ng/µl gene-specific crRNA, and 110 ng/µl of the oligo repair template. For injections using two gene-specific crRNAs, the injection mix included 0.25 µg/µl Cas9 protein (IDT), 100 ng/µl tracrRNA (IDT), 14 ng/µl dpy-10 crRNA, 21 ng/µl each gene-specific crRNA, and 110 ng/µl of each repair template.

The following strains were used for injection: *enri-2(cmp318)* and *enri-1(cmp320[enri-1::mCherry::2xHA])* into wild-type N2 strain. *nrde-3(cmp324[HK-AA]) and enri-1(cmp328)* into JMC237: *nrde-3(tor131[GFP::3xFLAG::nrde-3])* X. *rde-3/mut-2(cmp337)* into USC1615: *ego-1(cmp317[ego-1::degron]) I; ieSi38 [Psun-1::TIR1::mRuby::sun-1 3' UTR] IV; nrde-3(tor131[GFP::3xFLAG::nrde-3])* X. Following injection, F1 animals with the Rol phenotype were isolated and genotyped by PCR to identify heterozygous animals with the mutations of interest, then F2 animals were further singled out to identify homozygous mutant animals.

## RNAi assays

For RNAi experiment, control L4440 and *lgg-1* RNAi *E. coli* clones were sequenced verified and cultured at 37 °C for 16 hours, then RNAi bacteria were seeded on fresh RNAi plates. Two L4 animals were transferred to seeded RNAi plates and raised at 20 °C. When the progenies of the L4 animals turn into young adults (about 4 days), the embryos of young adult animals were imaged to assess somatic granule expression.

## Immunofluorescence imaging and quantification

For immunofluorescence, *C. elegans* were dissected in egg buffer containing 0.1% Tween-20 and fixed in 1% formaldehyde in egg buffer as described (*Phillips et al., 2009*). Samples were immunostained with anti-RNA Polymerase II CTD (phosphoSer2) at 1:500 (Abcam ab5095). The secondary antibody, anti-Rabbit IgG AlexaFluor 555 was used at 1:1000 (Thermo Fisher A21429). Animals were dissected at the adult stage (24 h post L4). Imaging was performed on a DeltaVision Elite microscope (GE Healthcare) using a 60 x N.A. 1.42 oil-immersion objective, data stacks were collected and deconvolution was performed using the SoftWoRx package. 25 optical sections, for a total 12.5 µm sample thickness, are presented as a maximum intensity projection. PSer2 signal intensity was calculated in ImageJ (version 1.53 a) using a minimum of 10 individual adult gonads.

## Live imaging

Live imaging of *C. elegans* embryos was performed in M9 buffer. Young embryos were obtained by dissecting gravid adult *C. elegans*, and old embryos were obtained by manually picking embryos laid on the NGM plate. Live imaging of *C. elegans* adult germline was performed in M9 buffer containing sodium azide to prevent movement. Day-one-adult *C. elegans* were obtained by manually picking L4s and leaving L4s at 20 °C for about 24 hr. Imaging was performed on a DeltaVision Elite microscope (GE Healthcare) using a 60 x N.A. 1.42 oil-immersion objective. Images were pseudocolored using Adobe Photoshop.

## Granule number quantification

Granule number quantification was performed in FIJI/ImageJ2 (version 2.9.0). At least 10 embryos were imaged on a DeltaVision Elite microscope with 37 optical sections of a total 22.20 µm sample thickness from the bottom of the sample. Images were deconvolved to eliminate backgrounds. Z stacks were opened using the 3D object counter plugin for FIJI, and the granule counting threshold for each image was manually adjusted to obtain the least background and most granules.

## Colocalization analysis

Quantitative colocalization analysis between different granules was performed in FIJI/ImageJ2 (version 2.9.0) using the Coloc2 package and 100 cell staged embryos imaged on a DeltaVision Elite microscope. At least 3 granules from each embryo, and at least 4 individual embryos for a total of at least 20 granules were used to calculate Pearson's R value.

## Western blot

Synchronized adult *C. elegans* were harvested (~72 hours at 20 °C after L1 arrest) and 200 adults were loaded per lane. Proteins were resolved on 4–12% Bis-Tris polyacrylamide gels (Thermo Fisher, NW04122BOX), transferred to nitrocellulose membranes (Thermo Fisher, LC2001), and probed with rat anti-HA-peroxidase 1:1000 (Roche 12013819001), mouse anti-FLAG 1:1000 (Sigma, F1804), or mouse anti-actin 1:10,000 (Abcam ab3280). Secondary HRP antibodies were purchased from Thermo Fisher. Unedited western blots are provided in the Source Data File.

## Small RNA library preparation and sequencing

For *C. elegans* embryo staging and collection, synchronized arrested L1s were grown on enriched peptone plates at 17 °C until the young adult stage. Adult *C. elegans* stage was monitored carefully under DeltaVision microscope by live imaging. For early embryo collection (≤ 100 cell), adult animals were washed off from plates with $H_2O$ and bleached as soon as the first animals had 1–4 eggs (around 68–70 hours depending on the strain and the incubator temperature). For late embryo collection (≥ 300 cell), adult animals were washed off from plates with $H_2O$ and bleached when about half of the worms had 1~6 eggs (around 70–72 hours depending on the strain and the incubator temperature). After bleaching, embryos were washed twice with M9 buffer, and filtered through 40 µm cell strainers (Fisherbrand Sterile Cell Strainers, 40 µm) twice to clear the residual worm body. To reach ≥ 300 cell stage for late embryo collection, embryos were additionally incubated in M9 buffer at 20 °C for 4.5 hr. Then embryos were washed once with IP buffer (50 mM Tris-Cl pH 7.5, 100 mM KCl, 2.5 mM MgCl2, 0.1% Nonidet P40 substitute) containing Protease Inhibitor (Thermo Fisher A32965). Embryos were kept on ice during washes to prevent further development. 500,000 embryos were collected for each replicate. Following washes, embryos were flash-frozen by placing tubes in a container with ethanol and dry ice. A small aliquot of embryos was examined on the Deltavision microscope to confirm the developmental stage immediately before freezing. Frozen embryos were stored at –80 °C until immunoprecipitation.

For immunoprecipitation followed by small RNA sequencing in embryos, ~500,000 synchronized embryos were sonicated with Fisher Sonifier 550 with a microtip (15 s on, 45 s off, 10% power, total 2 min on time). After sonication, insoluble particulate was removed by centrifugation at 21,000 × *g* for 30 min. Immunoprecipitation was performed using anti-FLAG Affinity Matrix (Sigma Aldrich, A2220). NRDE-3-bound RNAs were isolated using TRIzol reagent (Thermo Fisher, 15596018), followed by chloroform extraction and isopropanol precipitation. Small RNAs (18–30-nt) were size selected on homemade 10% Urea-polyacrylamide gels from total RNA samples. Small RNAs were treated with 5' RNA polyphosphatase (Epicenter RP8092H) and ligated to 3' pre-adenylated adapters with Truncated T4 RNA ligase (NEB M0373L). Small RNAs were then hybridized to the reverse transcription primer, ligated to the 5' adapter with T4 RNA ligase (NEB M0204L), and reverse transcribed with Superscript III (Thermo Fisher 18080–051). Small RNA libraries were amplified using Q5 High-Fidelity DNA polymerase (NEB M0491L) and size selected on a homemade 10% polyacrylamide gel. Library concentration was determined using the Qubit 1 X dsDNA HS Assay kit (Thermo Fisher Q33231) and quality was assessed using the Agilent BioAnalyzer. Libraries were sequenced on the Illumina NextSeq2000 (SE 75 bp reads) platform. Primer sequences are available in *Supplementary file 2*. Differentially expressed gene lists and gene lists used in this paper can be found in *Supplementary file 3*. Sequencing library statistics summary can be found in *Supplementary file 4*.

## Bioinformatic analysis

For small RNA libraries, sequences were parsed from adapters and quality filtered using FASTX-Toolkit (version 0.0.13) (*Hannon, 2010*). Filtered reads were mapped to the *C. elegans* genome, WS258, using Bowtie2 (version 2.5.0) (*Langmead and Salzberg, 2012*). Mapped reads were assigned to genomic features using featureCounts which is part of the Subread package (version 2.0.1) (*Liao et al., 2014*).

Differential expression analysis was performed using edgeR (3.40.2) (*Robinson et al., 2010*). To define gene lists from IP experiments, a twofold-change cutoff, an edgeR adjusted p-value of ≤0.05, and at least 10 RPM in the IP libraries were required to identify genes with significant changes in small RNA levels.

## Acknowledgements

We thank the members of the Phillips lab for helpful discussions and feedback on the manuscript, the labs of Julie Claycomb, John Kim, Heng-Chi Lee, and Mihail Sarov for generously providing strains, and the lab of Matt Michael for sharing the PSer2 antibody and advising on conditions for staining and quantification. Some strains were provided by the CGC, which is funded by NIH Office of Research Infrastructure Programs (P40 OD010440). Next generation sequencing was performed by the USC Molecular Genomics Core, which is supported by award number P30 CA014089 from the National Cancer Institute.

## Additional information

### Funding

| Funder | Grant reference number | Author |
| --- | --- | --- |
| National Institute of General Medical Sciences | R35 GM119656 | Carolyn Marie Phillips |

The funders had no role in study design, data collection and interpretation, or the decision to submit the work for publication.

### Author contributions
Shihui Chen, Conceptualization, Formal analysis, Investigation, Visualization, Writing – original draft, Writing – review and editing; Carolyn Marie Phillips, Conceptualization, Formal analysis, Supervision, Funding acquisition, Writing – original draft, Writing – review and editing

### Author ORCIDs
Shihui Chen https://orcid.org/0009-0006-3036-5475
Carolyn Marie Phillips https://orcid.org/0000-0002-6228-6468

Reviewer #1 (Public review): https://doi.org/10.7554/eLife.102226.3.sa1
Reviewer #2 (Public review): https://doi.org/10.7554/eLife.102226.3.sa2
Reviewer #3 (Public review): https://doi.org/10.7554/eLife.102226.3.sa3
Author response https://doi.org/10.7554/eLife.102226.3.sa4

## Additional files

### Supplementary files
Supplementary file 1. Strains used in this study.

Supplementary file 2. Oligonucleotides sequences used in this study.

Supplementary file 3. Small RNA enrichment in NRDE-3 immunoprecipitations.

Supplementary file 4. Sequencing library statistics.

MDAR checklist

### Data availability
The RNA sequencing data generated in this study are available through Gene Expression Omnibus (GEO) under accession code GSE273239. All data generated or analyzed during this study are included in the manuscript and supporting files; source data files have been provided for Figures 1,2,4,5,7 and 8.

The following dataset was generated:

| Author(s) | Year | Dataset title | Dataset URL | Database and Identifier |
|---|---|---|---|---|
| Chen S, Phillips CM | 2024 | Nuclear Argonaute protein NRDE-3 switches small RNA binding partners during embryogenesis coincident with the formation of SIMR granules | https://www.ncbi.nlm.nih.gov/geo/query/acc.cgi?acc=GSE273239 | NCBI Gene Expression Omnibus, GSE273239 |

The following previously published datasets were used:

| Author(s) | Year | Dataset title | Dataset URL | Database and Identifier |
|---|---|---|---|---|
| Quarato P, Singh M, Cornes E, Li B, Didier C, Bourdon L, Mueller F, Cecere G | 2021 | Germline inherited small RNAs facilitate the clearance of untranslated maternal mRNAs in *C. elegans* embryos (RNA-seq) | https://www.ncbi.nlm.nih.gov/geo/query/acc.cgi?acc=GSE146057 | NCBI Gene Expression Omnibus, GSE146057 |
| Quarato P, Singh M, Cornes E, Li B, Didier C, Bourdon L, Mueller F, Cecere G | 2021 | Germline inherited small RNAs facilitate the clearance of untranslated maternal mRNAs in *C. elegans* embryos | https://www.ncbi.nlm.nih.gov/geo/query/acc.cgi?acc=GSE146062 | NCBI Gene Expression Omnibus, GSE146062 |
| Padeken J, Methot S, Zeller P, Delaney C, Kalck V, Gasser SM | 2020 | Two parallel pathways recruit the H3K9me3 HMT in somatic cells, requiring the Argonaut NRDE-3, or the MBT-domain protein, LIN-61 (ChIP-seq) | https://www.ncbi.nlm.nih.gov/geo/query/acc.cgi?acc=GSE156548 | NCBI Gene Expression Omnibus, GSE156548 |
| Seroussi U, Lugowski A, Wadi L, Lao RX, Willis AR, Zhao W, Sundby AE, Charlesworth AG, Reinke AW, Claycomb JM | 2022 | A Comprehensive Survey of *C. elegans* Argonaute Proteins Reveals Organism-wide Gene Regulatory Networks and Functions | https://www.ncbi.nlm.nih.gov/geo/query/acc.cgi?acc=GSE208702 | NCBI Gene Expression Omnibus, GSE208702 |

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
