## [Editor Report · eLife Assessment]

The study by Chen and Phillips provides evidence for a dynamic switch in the small RNA repertoire of the Argonaute protein NRDE-3 during embryogenesis in *C. elegans*. The work is supported by **convincing** experimental data, shedding light on RNA regulation during development. While the functional relevance of this process warrants further investigation, this study provides **valuable** insights into small RNA pathways with broader implications for developmental biology and gene regulation in other systems.

---

## [Referee Report · Reviewer #1 (Public review)]

Summary:

Chen and Phillips describe the dynamic appearance of cytoplasmic granules during embryogenesis analogous to SIMR germ granules, and distinct from CSR-1-containing granules, in the *C. elegans* germline. They show that the nuclear Argonaute NRDE-3, when mutated to abrogate small RNA binding, or in specific genetic mutants, partially colocalizes to these granules along with other RNAi factors, such as SIMR-1, ENRI-2, RDE-3, and RRF-1. Furthermore, NRDE-3 RIP-seq analysis in early vs. late embryos is used to conclude that NRDE-3 binds CSR-1-dependent 22G RNAs in early embryos and ERGO-1-dependent 22G RNAs in late embryos. These data lead to their model that NRDE-3 undergoes small RNA substrate ‘switching’ that occurs in these embryonic SIMR granules and functions to silence two distinct sets of target transcripts - maternal, CSR-1 targeted mRNAs in early embryos and duplicated genes and repeat elements in late embryos.

Strengths:

The identification and function of small RNA-related granules during embryogenesis is a poorly understood area and this study will provide the impetus for future studies on the identification and potential functional compartmentalization of small RNA pathways and machinery during embryogenesis.

Weaknesses:

(1) The authors acknowledge the following issue that loss of SIMR granules have no significant impact on NRDE-3 small RNA loading weakens the functional relevance of these structures. However, this point is clearly discussed and, as they note in their Discussion, it is entirely possible that these embryonic granules may be ‘incidental condensates.’

---

## [Referee Report · Reviewer #2 (Public review)]

Summary:

NRDE-3 is a nuclear WAGO-clade Argonaute that, in somatic cells, binds small RNAs amplified in response to the ERGO-class 26G RNAs that target repetitive sequences. This manuscript reports that, in the germline and early embryos, NRDE-3 interacts with a different set of small RNAs that target mRNAs. This class of small RNAs were previously shown to bind to a different WAGO-clade Argonaute called CSR-1, which is cytoplasmic unlike nuclear NRDE-3. The switch in NRDE-3 specificity parallels recent findings in Ascaris where the Ascaris NRDE homolog was shown to switch from sRNAs that target repetitive sequences to CSR-class sRNAs that target mRNAs.

The manuscript also correlates the change in NRDE-3 specificity with the appearance in embryos of cytoplasmic condensates that accumulate SIMR-1, a scaffolding protein that the authors previously implicated in sRNA loading for a different nuclear Argonaute HRDE-1. By analogy, and through a set of corelative evidence, the authors argue that SIMR foci arise in embryogenesis to facilitate the change in NRDE-3 small RNA repertoire. The paper presents lots of data that beautifully documents the appearance and composition of the embryonic SIMR-1 foci, including evidence that a mutated NRDE-3 that cannot bind sRNAs accumulate in SIMR-1 foci in SIMR-1-dependent fashion.

---

## [Referee Report · Reviewer #3 (Public review)]

Summary:

Chen and Phillips present intriguing work that extends our view on the *C. elegans* small RNA network significantly. While the precise findings are rather *C. elegans* specific there are also messages for the broader field, most notably the switching of small RNA populations bound to an argonaute, and RNA granules behavior depending on developmental stage. The work also starts to shed more light on the still poorly understood role of the CSR-1 argonaute protein and supports its role in the decay of maternal transcripts. Overall, the work is of excellent quality, and the messages have a significant impact.

Strengths:

Compelling evidence for major shift in activities of an argonaute protein during development, and implications for how small RNAs affect early development. Very balanced and thoughtful discussion.

Weaknesses:

The switch between maternal and zygotic NRDE-3 remains unaddressed

---

## [Author Response]

The following is the authors’ response to the original reviews.

**Reviewer #1 (Public review):**
Summary:Chen and Phillips describe the dynamic appearance of cytoplasmic granules during embryogenesis analogous to SIMR germ granules, and distinct from CSR-1-containing granules, in the *C. elegans* germline. They show that the nuclear Argonaute NRDE-3, when mutated to abrogate small RNA binding, or in specific genetic mutants, partially colocalizes to these granules along with other RNAi factors, such as SIMR-1, ENRI-2, RDE-3, and RRF-1. Furthermore, NRDE-3 RIP-seq analysis in early vs. late embryos is used to conclude that NRDE-3 binds CSR-1-dependent 22G RNAs in early embryos and ERGO-1dependent 22G RNAs in late embryos. These data lead to their model that NRDE-3 undergoes small RNA substrate ‘switching’ that occurs in these embryonic SIMR granules and functions to silence two distinct sets of target transcripts - maternal, CSR-1 targeted mRNAs in early embryos and duplicated genes and repeat elements in late embryos.Strengths:The identification and function of small RNA-related granules during embryogenesis is a poorly understood area and this study will provide the impetus for future studies on the identification and potential functional compartmentalization of small RNA pathways and machinery during embryogenesis.Weaknesses:(1) While the authors acknowledge the following issue, their finding that loss of SIMR granules has no apparent impact on NRDE-3 small RNA loading puts the functional relevance of these structures into question. As they note in their Discussion, it is entirely possible that these embryonic granules may be ‘incidental condensates.’ It would be very welcomed if the authors could include some evidence that these SIMR granules have some function; for example, does the loss of these SIMR granules have an effect on CSR-1 targets in early embryos and ERGO-1-dependent targets in late embryos?

We appreciate reviewer 1’s concern that we do not provide enough evidence for the function of the SIMR granules. As suggested, we examined the NRDE-3 bound small RNAs more deeply, and we do observe a slight but significant increased CSR-class 22G-RNAs binding to NRDE-3 in late embryos of *simr-1* and *enri-2* mutants (see below, right). We hypothesize that this result could be due to a slower switch from CSR to ERGO 22G-RNAs in the absence of SIMR granules. We added these data to Figure 6G.

(2) The analysis of small RNA class ‘switching’ requires some clarification. The authors re-define ERGO1-dependent targets in this study to arrive at a very limited set of genes and their justification for doing this is not convincing. What happens if the published set of ERGO-1 targets is used?

As we mentioned in the manuscript, we initially attempted to use the previously defined ERGO targets. However, the major concern is fewer than half the genes classified as ERGO targets by Manage *et al.* and Fischer *et al.* overlap with one another (Figure 6—figure supplement 1D and below). We reason this might because the gene sets were defined as genes that lose small RNAs in various ERGO pathway mutants and because different criteria were used to define the lists as discussed in the manuscript (lines 471-476). As a result, some of the previously defined ERGO target genes may actually be indirect targets of the pathway. Here we focus on genes targeted by small RNAs enriched in an ERGO pathway Argonaute IP, which should be more specific.

In this manuscript, we are interested specifically in the ERGO targets bound by NRDE-3, thus we utilized the IP-small RNA sequencing data from young adult animals (Seroussi et al, 2023), to define a new ERGO list. We are confident about this list because (1) Most of our new ERGO genes overlap with the overlap between ERGO-Manage and ERGO-Fischer list (see Figure 6—figure supplement 1D in our manuscript and below). (2) We observed the most significant decrease of small RNA levels and increase of mRNA levels in the *nrde-3* mutants using our newly defined list (see Figure 6—figure supplement 1E-F in our manuscript).

To further address reviewer 1’s concern about whether the data would look significantly different when using the ERGO-Manage and ERGO-Fischer lists, we made new scatter plots shown in Author response image 1 panels A-C below (ERGO-Manage – purple, ERGO-Fischer- yellow, and the overlap - yellow with purple ring). We found that the small switching pattern of NRDE-3 is consistent with our newly defined list, particularly if we look at the overlap of ERGO-Manage and ERGO-Fischer list (Author response image 1 panels D-F below, red).

**Author response image 1. sa4fig1:** 

Further, the NRDE-3 RIP-seq data is used to conclude that NRDE-3 predominantly binds CSR-1 class 22G RNAs in early embryos, while ERGO-1-dependent 22G RNAs are enriched in late embryos. (a) The relative ratios of each class of small RNAs are given in terms of unique targets. What is the total abundance of sequenced reads of each class in the NRDE-3 IPs?

To address the reviewer’s question about the total abundance of sequenced reads of each class in the NRDE-3 IPs: Author response image 2 panel A-B below show the total RPM of CSR and ERGO class sRNAs in inputs and IPs at different stages. Focusing on late embryos, the total abundance of ERGO-dependent sRNAs is similar to CSR-class sRNAs in input, while much higher in IP, indicating an enrichment of ERGO-dependent 22G-RNAs in NRDE-3 consistent with our log2FC (IP vs input) in Figure 6B. This data supports our conclusion that NRDE-3 preferentially binds to ERGO targets in late embryos.

**Author response image 2. sa4fig2:** 

b) The ‘switching’ model is problematic given that even in late embryos, the majority of 22G RNAs bound by NRDE-3 is the CSR-1 class (Figure 5D).

It is important to keep in mind the difference in the total number of CSR target genes (3834) and ERGO target genes (119). The pie charts shown in Figure 6D are looking at the total proportion of the genes enriched in the NRDE-3 IP that are CSR or ERGO targets. For the NRDE-3 IP in late embryos, that would be 70/119 (58.8%) of ERGO targets are enriched, while 172/3834 (4.5%) of CSR targets are enriched. These data are also supported by the RPM graphs shown in Author response image 2 panels A-B above, which show that the majority of the small RNA bound by NRDE-3 in late embryos are ERGO targets. Nonetheless, NRDE-3 still binds to some CSR targets shown as Figure 6D and panel B, which may be because the amount of CSR-class 22G-RNAs is reduced gradually across embryonic development as the maternally-deposited NRDE-3 loaded with CSR-class 22G-RNAs is diluted by newly transcribed NRDE-3 loaded with ERGOdependent 22G-RNAs (lines 857-862).

c) A major difference between NRDE-3 small RNA binding in eri-1 and simr-1 mutants appears to be that NRDE-3 robustly binds CSR-1 22G RNAs in eri-1 but not in simr-1 in late embryos. This result should be better discussed.

In the *eri-1* mutant, we hypothesize that NRDE-3 robustly binds CSR-class 22G-RNAs because ERGOclass 22G-RNAs are not synthesized during mid-embryogenesis, so either NRDE-3 is unloaded (in granule at 100-cell stage in Figure 2A) or mis-loaded with CSR-class 22G-RNAs (in the nucleus at 100cell stage in Figure 2A). We don’t have a robust method to address the proportion of loaded vs. unloaded NRDE-3 so it is difficult to address the degree to which NRDE-3 is misloaded in the *eri-1* mutant. In the *simr-1* mutant, both classes of small RNAs are present and NRDE-3 is still preferentially loaded with ERGO-dependent 22G-RNAs, though we do see a subtle increase in association with CSR-class 22GRNAs. These data could suggest a less efficient loading of NRDE-3 with ERGO-dependent 22G-RNAs, but we would need more precise methods to address the loading dynamics in the *simr-1* mutant.

(3) Ultimately, if the switching is functionally important, then its impact should be observed in the expression of their targets. RNA-seq or RT-qPCR of select CSR-1 and ERGO-1 targets should be assessed in nrde-3 mutants during early vs late embryogenesis.

The function of NRDE-3 at ERGO targets has been well studied (Guang *et al*, 2008) and is also assessed in our H3K9me3 ChIP-seq analysis in Figure 7E where, in mixed staged embryos, H3K9me3 level on ERGO targets (labeled as ‘*NRDE-3 targets in young adults’*) is reduced significantly in the *nrde-3* mutant.

To understand the function of NRDE-3 binding on CSR targets in early embryos, we attempted to do RTqPCR, smFISH, and anti-H3K9me3 CUT&Tag-seq on early embryos, and we either failed to obtain enough signal or failed to detect any significant difference (data not shown). We additionally tested the possibility that NRDE-3 functions with CSR-class 22G-RNAs in oocytes. We present new data showing that NRDE-3 represses RNA Pol II in oocytes to promote global transcriptional repression at the oocyteto-embryo transition, we now included these data in Figure 8.

**Reviewer #2 (Public review):**
Summary:NRDE-3 is a nuclear WAGO-clade Argonaute that, in somatic cells, binds small RNAs amplified in response to the ERGO-class 26G RNAs that target repetitive sequences. This manuscript reports that, in the germline and early embryos, NRDE-3 interacts with a different set of small RNAs that target mRNAs. This class of small RNAs was previously shown to bind to a different WAGO-clade Argonaute called CSR1, which is cytoplasmic, unlike nuclear NRDE-3. The switch in NRDE-3 specificity parallels recent findings in Ascaris where the Ascaris NRDE homolog was shown to switch from sRNAs that target repetitive sequences to CSR-class sRNAs that target mRNAs.The manuscript also correlates the change in NRDE-3 specificity with the appearance in embryos of cytoplasmic condensates that accumulate SIMR-1, a scaffolding protein that the authors previously implicated in sRNA loading for a different nuclear Argonaute HRDE-1. By analogy, and through a set of corelative evidence, the authors argue that SIMR foci arise in embryogenesis to facilitate the change in NRDE-3 small RNA repertoire. The paper presents lots of data that beautifully documents the appearance and composition of the embryonic SIMR-1 foci, including evidence that a mutated NRDE-3 that cannot bind sRNAs accumulates in SIMR-1 foci in a SIMR-1-dependent fashion.Weaknesses:The genetic evidence, however, does not support a requirement for SIMR-1 foci: the authors detected no defect in NRDE-3 sRNA loading in simr-1 mutants. Although the authors acknowledge this negative result in the discussion, they still argue for a model (Figure 7) that is not supported by genetic data. My main suggestion is that the authors give equal consideration to other models - see below for specifics.

We appreciate reviewer 2’s comments on the genetic evidence for the function of SIMR foci. A similar concern was also brought up by reviewer 1. By re-examining our sequencing data, we found that there is a modest but significant increase in NRDE-3 association with CSR-class sRNAs in *simr-1* and *enri-2* mutants in late embryos. We believe that this data supports our model that SIMR-1 and ENRI-2 are required for an efficient switch of NRDE-3 bound small RNAs. Please refer our response to the reviewer 1 - point (1), and Figure 6G in the updated manuscript.

**Reviewer #3 (Public review):**
Summary:Chen and Phillips present intriguing work that extends our view on the *C. elegans* small RNA network significantly. While the precise findings are rather *C. elegans* specific there are also messages for the broader field, most notably the switching of small RNA populations bound to an argonaute, and RNA granules behavior depending on developmental stage. The work also starts to shed more light on the still poorly understood role of the CSR-1 argonaute protein and supports its role in the decay of maternal transcripts. Overall, the work is of excellent quality, and the messages have a significant impact.Strengths:Compelling evidence for major shift in activities of an argonaute protein during development, and implications for how small RNAs affect early development. Very balanced and thoughtful discussion.Weaknesses:Claims on col-localization of specific 'granules' are not well supported by quantitative data

We have now included zoomed images of individual granules to better show the colocalization in Figure 4 and Figure 4—figure supplement 1, and performed Pearson’s colocalization analysis between different sets of proteins in Figure 4B.

**Reviewer #2 (Recommendations for the authors):**
- The manuscript is very dense and the gene names are not helpful. For example, the authors mention ERGO-1 without clarifying the type of protein, etc. I suggest the authors include a figure to go with the introduction that describes the different classes of primary and secondary sRNAs, associated Argonautes, and other accessory proteins. Also include a table listing relevant gene names, protein classes, main localizations, and proposed functions for easy reference by the readers.

We agree that the genes names in different small RNA pathways are easily confused. We added a diagram and table in Figure 1—figure supplement 1 depicting the ERGO/NRDE and CSR pathways and added clarification about the ERGO/NRDE-3 pathway in the text from line 126-128.

- Line 424 - the wording here and elsewhere seems to imply that SIMR-1 and ENRI-2, although not essential, contribute to NRDE-3 sRNA loading. The sequencing data, however, do not support this - the authors should be clearer on this. If the authors believe there are subtle but significant differences, they should show them perhaps by adding a panel in Figure 5 that directly compares the NRDE-3 IPs in wildtype versus simr-1 mutants. Figure 5H however does not support such a requirement.

As brought up by reviewer 1, we do not see difference in binding of ERGO-dependent sRNA in *simr-1* mutant in late embryos. We do, however, see a modest, but significant, increase of CSR-sRNAs bound by NRDE-3 in *simr-1* and *enri-2* mutants, which we hypothesize could be due to a less efficient loading of ERGO-dependent 22G-RNAs by NRDE-3. The updated data are now in Figure 6G. We have also edited the text and model figure to soften these conclusions.

- Condensates of PGL proteins appear at a similar time and place (somatic cells of early embryos) as the embryonic SIMR-1 foci. The PGL foci correspond to autophagy bodies that degrade PGL proteins. Is it possible that SIMR-1 foci also correspond to degradative structures? The possibility that SIMR-1 foci are targeted for autophagy and not functional would fit with the finding that simr-1 mutants do not affect NRDE-3 loading in embryos.

We appreciate reviewer 2’s comments on possibility of SIMR granules acting as sites for degradation of SIMR-1 and NRDE-3. We think this is not the case for the following reasons: (1) if SIMR granules are sites of autophagic degradation, then we would expect that embryonic SIMR granules in somatic cells, like PGL granules, should only be observed in autophagy mutants; however we see them in wild-type embryos (2) we would not expect a functional Tudor domain to be required for granule localization; however in Figure 1—figure supplement 2B, we show that a point mutation in the Tudor domain of SIMR-1 abrogates SIMR granule formation, and (3) if NRDE-3(HK-AA) is recruited to SIMR granules for degradation while wild-type NRDE-3 is cytoplasmic, then NRDE-3(HK-AA) should shows a significantly reduced protein level comparing to wild-type NRDE-3. In the western blot in Figure 2—figure supplement 1B, NRDE-3 and NRDE-3(HK-AA) protein levels are similar, indicating that NRDE-3(HK-AA) is not degraded despite being unloaded. This is in contrast to what we have observed previously for HRDE-1, which is degraded in its unloaded state. If SIMR-1 played a role directly in promoting degradation of NRDE-3(HK-AA), we would similarly expect to see a change in NRDE-3 or NRDE-3(HK-AA) expression in a *simr-1* mutant. We performed western blot and did not observe a significant change in protein expression for NRDE-3 (Figure 3—figure supplement 1A).

Although under wild-type conditions, SIMR granules do not appear to be sites of autophagic degradation, upon treatment with *lgg-1* (an autophagy protein) RNAi, we found that SIMR-1, as well as many other germ granule and embryonic granule-localized proteins, increase in abundance in late embryos. This data demonstrates that ZNFX-1, CSR-1, SIMR-1, MUT-2/RDE-3, RRF-1, and unloaded NRDE-3 are removed by autophagic degradation similar to what have been shown previously for PGL-1 proteins (Zhang et al, 2009, Cell). We added these data to Figure 5. It is important to emphasize, however, that the timing of degradation differs for each granule assayed (Lines 447-450), indicating that there must be multiple waves of autophagy to selectively degrade subsets of proteins when they are no longer needed by the embryo.

- The observation that an NRDE-3 mutant that cannot load sRNAs localizes to SIMR-1 foci does not necessarily imply that wild-type unloaded NRDE-3 would also localize there. Unless the authors have additional data to support this idea, the authors should acknowledge that this hypothesis is speculative. In fact, why does cytoplasmic NRDE-3 not localize to granules in the rde-3;ego-1degron strain shown in Figure 6B?? Is it possible that the NRDE-3 mutant accumulates in SIMR-1 foci because it is unfolded and needs to be degraded?

We believe that wild-type NRDE-3 also localize to SIMR foci when unloaded. This is supported by the localization of wild-type NRDE-3 in *eri-1* and *rde-3* mutants, where a subset of small RNAs are depleted. Wild-type NRDE-3 localizes to both somatic SIMR-1 granules and the nucleus, depending on embryo stage (Figure 2A, Figure 2—figure supplement 1C). The granule numbers in *eri-1* and *rde-3* mutants are less than the *nrde-3(HK-AA)* mutant, consistent with the imaging data that NRDE-3 only partially localize to somatic granule (Figure 2A – 100-cell stage).

In the *rde-3; ego-1* double mutant, the embryos have severe developmental defect: they cannot divide properly after 4-8 cell stage and exhibit morphology defects after that stage. In wild-type, SIMR foci does not appear until around 8-28-cell stage (shown in Figure 1C), so we believe that cytoplasmic NRDE-3 does not localize to foci in the double mutant is because of the timing.

- The authors propose that NRDE-3 functions in nuclei to target mRNAs also targeted in the cytoplasm by CSR-1. If so, how do they propose that NRDE-3 might do this since little transcription occurs in oocytes/early embryos?? Are the authors suggesting that NRDE-3 targets germline genes for silencing specifically at the times that zygotic transcription comes back on, or already in maturing oocytes? Is the transcription of most CSR-1 targets silenced in early embryos??

We appreciate the suggestions to check the function of NRDE-3 in oocytes. We tested this possibility and found it to be correct. NRDE-3 functions in oocytes for transcriptional repression by inhibiting RNA Pol II elongation. We added these data to Figure 8. We also attempted to do RT-qPCR, smFISH, and antiH3K9me3 Cut&Tag-seq on early embryos to further test the hypothesis that NRDE-3 acts with CSR-class 22G-RNAs in early embryos, but we either failed to obtain enough signal or failed to detect any significant difference (data not shown). Therefore, we think that the primary role for NRDE-3 bound to CSR-class 22G-RNAs may be for global transcriptional repression of oocytes prior to fertilization.

- Line 684-686: ‘In summary, this work investigating the role of SIMR granules in embryos, together with our previous study of SIMR foci in the germline (Chen and Phillips 2024), has identified a new mechanism for small RNA loading of nuclear Argonaute proteins in *C. elegans’*. This statement appears overstated/incorrect since there is no evidence that SIMR-1 foci are required for sRNA loading of NRDE3. The authors should emphasize other models, as suggested above.

We have revised the text on line 869-871 to emphasize that SIMR granule regulate the localization of nuclear Argonaute proteins, rather than suggesting a direct role on controlling small RNA loading. We also edit the title, text, and legend for our model in Figure 9.

**Reviewer #3 (Recommendations for the authors):**
Issues to be addressed:- The authors show a switch in 22G RNA binding by NRDE-3 during embryogenesis. While the data is convincing, it would be great if it could be tested if the preferred NRDE-3 replacement model is indeed correct. This could be done relatively easily by giving NRDE-3 a Dendra tag, allowing one to colour-switch the maternal WAGO-3 pool before the zygotic pool comes up. Such data would significantly enhance the manuscript, as this would allow the authors to follow the fate of maternal NRDE-3 more precisely, perhaps identifying a period of sharp decline of maternal NRDE-3.

We think the NRDE-3 Dendra tag experiment suggested by the reviewer is a clever approach and we will consider generating this strain in the future. However, we feel that optimization of the color-switching tag between the maternal germline and the developing embryos is beyond the scope of this manuscript. To partially address the question about NRDE-3 fate during embryogenesis, we examined the single-cell sequencing data of *C. elegans* embryos from 1-cell to 16-cell stage (Tintori *et al*, 2016, Dev Cell; Visualization tool from John I Murray lab), as shown in Author response image 3 Panel A below, NRDE-3 transcript level increases as embryo develops, indicating that zygotic NRDE-3 is being actively expressed starting very early in development. We hypothesize that maternal NRDE-3 will either be diluted as the embryo develops or actively degraded during early embryogenesis.

**Author response image 3. sa4fig3:** 

- Figure 3A: * should mark PGCs, but this seems incorrect. At the 8-cell stage there still is only one PGC (P4), not two, and at 100 cells there are only two, not three germ cells. Also, the identification of PGCs with a maker (PGL for instance) would be much more convincing.

We apologize for the confusion in Figure 3A. We changed the figure legend to clarify that the * indicate nuclear NRDE-3 localization in somatic cells for 8- and 100-cell stage embryos rather than the germ cells.

- Overall, the authors should address colocalization more robustly. In the current manuscript, just one image is provided, and often rather zoomed-out. How robust are the claims on colocalization, or lack thereof? With the current data, this cannot be assessed. Pearson correlation, combined with line-scans through a multitude of granules in different embryos will be required to make strong claims on colocalization. This applies to all figures (main and supplement) where claims on different granules are derived from.

We thank reviewer 3 for this important suggestion. To better address the colocalization, we included insets of individual granules in Figure 2D and Figure 4. We also performed colocalization analysis by calculating the Pearson’s R value between different groups of proteins in Figure 4B, to highlight that SIMR-1 colocalizes with ENRI-2, NRDE-3(HK-AA), RDE-3, and RRF-1, while CSR-1 colocalizes with EGO-1.

For the proteins that lack colocalization in Figure 4—figure supplement 1, we also added insets of individual granules. Additionally, we included a new set of panels showing SIMR-1 localization compared to tubulin::GFP (Figure 4—figure supplement 1I) in response to a recent preprint (Jin et al, 2024, BioRxiv), which finds NRDE-3 (expressed under a *mex-5* promoter) associating with pericentrosomal foci and the spindle in early embryos. We do not see SIMR-1 (or NRDE-3, data not shown) at centrosomes or spindles in wild-type conditions but made a similar observation for SIMR-1 in a *mut-16* mutant (Figure 4E). All of the localization patterns were examined on at least 5 individual 100-cell staged embryos with same localization pattern.

- Figure 7: Its title is: Function of cytoplasmic granules. This is a much stronger statement than provided in the nicely balanced discussion. The role of the granules remains unclear, and they may well be just a reflection of activity, not a driver. While this is nicely discussed in the text, figure 7 misses this nuance. For instance, the title suggests function, and also the legend uses phrases like 'recruited to granule X'. If granules are the results of activity, 'recruitment' is really not the right way to express the findings. The nuance that is so nicely worded in the discussion should come out fully in this figure and its legend as well.

We have changed the title of Figure 7 (now Figure 9) to ‘Model for temporally- and developmentallyregulated NRDE-3 function’ to deemphasize the role of the granules and to highlight the different functions of NRDE-3. Similarly, we have rephrased the text in the figure and legend and add a some details about our new results.

Minor:Typo: line 663 Acaris

We corrected the typo.